# Twin pair analysis uncovers links between DNA methylation, mitochondrial DNA quantity and obesity

Aino Heikkinen [1,2] ✉, Vivienne F. C. Esser [3], Seung Hyuk T. Lee [4], Sara Lundgren[1], Antti Hakkarainen [5], Jesper Lundbom[5,6], Juho Kuula[5,7], Per-Henrik Groop [8,9,10], Sini Heinonen[11,12], Sergio Villicaña [13], Jordana T. Bell [13], Alice Maguolo [14], Emma Nilsson[14], Charlotte Ling [14], Allan Vaag[15,16,17], Päivi Pajukanta [4,18,19], Jaakko Kaprio [1], Kirsi H. Pietiläinen [11,20], Shuai Li [3,21] & Miina Ollikainen [1,2] ✉

Alterations in mitochondrial metabolism in obesity may indicate disrupted communication between mitochondria and nucleus, and DNA methylation may influence this interplay. Here, we leverage data from the Finnish Twin Cohort study subcohort ($n = 173$; 86 full twin pairs, 1 singleton), including comprehensive measurements of obesity-related outcomes, mitochondrial DNA quantity and nuclear DNA methylation levels in adipose and muscle tissue, to identify one CpG at *SH3BP4* significantly associated with mitochondrial DNA quantity in adipose tissue (FDR < 0.05). We also show that *SH3BP4* methylation correlates with its gene expression. Additionally, we find that 14 out of the 35 obesity-related traits display significant associations with both *SH3BP4* methylation and mitochondrial DNA quantity in adipose tissue. We use data from TwinsUK and the Scandinavian T2D-discordant monozygotic twin cohort, to validate the observed associations. Further analysis using ICE FAL-CON suggests that mitochondrial DNA quantity, insulin sensitivity and certain body fat measures are causal to *SH3BP4* methylation. Examining mitochondrial DNA quantity and obesity-related traits suggests causation from mitochondrial DNA quantity to obesity, but unmeasured within-individual confounding cannot be ruled out. Our findings underscore the impact of mitochondrial DNA quantity on DNA methylation and expression of the *SH3BP4* gene within adipose tissue, with potential implications for obesity.

The significance of mitochondria as a primary energy source for cell growth and survival is indisputable. While mitochondria contain their own circular 16.6 kb genome, most of the genes required for mitochondrial function are encoded in the nuclear DNA. Therefore, cells require synchronized communication between the nucleus and mitochondria, known as mitonuclear communication, to adapt to changing metabolic demands. This intricate interplay can be subject to

regulation by epigenetic mechanisms[1,2], including DNA methylation[3], which may influence the activity of gene expression. Consequently, the importance of DNA methylation for mitochondrial function has been highlighted in the literature[4].

The existing body of literature supports a bidirectional relationship between mitochondrial function and nuclear DNA methylation. On one hand, different mitochondrial haplotypes and variants can lead

to differences in DNA methylation[5–8] via numerous metabolic pathways, such as the methionine cycle and the production of methyl groups[4]. On the other hand, nuclear DNA methylation may impact mitochondrial metabolism by regulating mitochondrial-associated gene expression[9–12]. The term 'mitochondrial metabolism' is used here as an umbrella term for any measurable trait associated with mitochondrial metabolism, such as mtDNA quantity (mtDNAq), copy number, biogenesis, dynamics, and OXPHOS (oxidative phosphorylation) activity.

The vital role of mitochondria in cellular energy metabolism has placed them in the center of interest in human traits and diseases with metabolic symptoms, including obesity.

Obesity is characterized by excess body fat, which can lead to a range of systemic metabolic disturbances, and it has been previously associated with compromised mitochondrial biogenesis and OXPHOS capacity[13,14]. Furthermore, obesity is a highly heterogeneous trait, and the precise molecular phenotypes contributing to the altered mitochondrial metabolism remain unclear. In addition, the exact tissue-specific roles of mitochondria in obesity are not well documented, although it has been shown that adipose tissue metabolism may be more affected by the acquired weight than muscle tissue[14].

The research on the role of DNA methylation underlying obesity-associated decline in mitochondrial metabolism remains scarce. A recent study showed that mtDNAq influences cardiovascular disease and mortality through changes in DNA methylation[15]. DNA methylation profiles of adipocyte progenitor cells of individuals with obesity have also been linked to mitochondrial metabolism[16]. Despite these insights, the causal pathways and comprehensive characterization of the orchestrated effects of DNA methylation and mitochondrial decline in obesity and obesity-related phenotypes remain elusive.

Here, we aimed to identify DNA methylation sites associated with differences in mtDNAq in two primary tissues affected by obesity, namely subcutaneous adipose tissue (SAT) and skeletal muscle. In addition, we explored whether these methylation sites are linked to a comprehensive range of obesity-related outcomes and other related phenotypes, including several anthropometric and body composition measures, clinical biomarkers, physical activity, and biological aging[17]. Finally, we explored the potential causal relationships between mtDNAq, DNA methylation, and obesity-related measures with a method called Inference about Causation from Examination of FAmilial CONfounding (ICE FALCON)[18].

## Results

A total of 173 individuals participated in the study, comprising 81 complete MZ twin pairs, 5 dizygotic (DZ) twin pairs, and 1 MZ singleton without available co-twin data (Table 1 and Supplementary Table 1). By singletons, we refer to twin individuals who lack their co-twin's data in the dataset or analysis. The DZ pairs and singletons were included in the first part of the study, i.e., the EWAS, whereas the ICE FALCON causation analysis was conducted first for complete pairs of MZ twins only and then using both MZ and DZ twin pairs. The age range of the study cohort spanned from 23 to 70 years old, with females accounting for 59% of the cohort (Table 1). The mean body mass index (BMI) and fat percentage were 29.2 kg/m² and 36.6%, respectively, indicating that the sample is predominantly with overweight. In addition, average fasting glucose levels (5.6 mmol/l) and HOMA-IR measures (2.1 units) were slightly elevated, which suggests a reduced insulin sensitivity. Twenty-two participants were diagnosed with type 2 diabetes (T2D). The main analysis strategy is described in Fig. 1, while Supplementary Table 2 summarizes the epigenetic age acceleration estimates of the study participants. Furthermore, the MZ twin pairs with adipose tissue data (n = 71 pairs) used in the causal inference analysis are presented in Supplementary Table 3.

## Table 1 | Participant characteristics

| N (twin pairs, singletons) | 173 (86, 1) |
|---|---|
| MZ / DZ (full twin pairs) | 163 (81) / 10 (5) |
| Age, mean (range) | 45.7 (22.8–69.3) |
| Female, % | 59.0 |
| Smoking, % | Never 39.3<br>Current 30.1<br>Former 30.6 |
| **Obesity-related measures, Mean (range) SD** | |
| Weight (kg) | 83.7 (48.7–143.6) 18.3 |
| BMI (kg/m²) | 29.2 (19.7–45.9) 5.8 |
| Waist circumference (cm) | 97.6 (65.2–144.3) 15.7 |
| WHR | 0.9 (0.8–1.1) 0.1 |
| Fat (%) | 36.6 (8.9–57.3) 9.6 |
| Fat (kg) | 31.9 (7.4–66.0) 13.0 |
| Fat-free mass (kg) | 49.8 (29.5–76.7) 10.8 |
| Intra-abdominal fat (cm³) | 1015 (152–3950) 870 |
| Subcutaneous fat (cm³) | 4582 (827–15,129) 2830 |
| Adipocyte volume (mm³) | 539 (123–1131) 218 |
| Liver fat (%) | 2.7 (0.1–22.4) 4.0 |
| Total cholesterol (mmol/l) | 4.8 (2.7–7.7) 0.9 |
| HDL (mmol/l) | 1.5 (0.5–3) 0.5 |
| LDL (mmol/l) | 3 (1–5.1) 0.8 |
| Triglycerides (mmol/l) | 1 (0.3–5.9) 0.6 |
| hsCRP (mg/l) | 2.3 (0.1–9.7) 2.3 |
| Adipsin (µg/ml) | 1.2 (0.7–1.7) 0.2 |
| Adiponectin (ng/ml) | 3320 (1310–7710) 1408 |
| ALAT (U/l) | 29.3 (14–127) 20.1 |
| ASAT (U/l) | 29.2 (7–131) 14.3 |
| Systolic blood pressure (mmHg) | 133 (97–195) 19 |
| Diastolic blood pressure (mmHg) | 80 (48–110) 78 |
| Fasting glucose (mmol/l) | 5.6 (4.0–16.1) 1.1 |
| Fasting insulin (mU/l) | 8.1 (0.9–34.3) 5.9 |
| HOMA-IR | 2.1 (0.2–9.9) 1.8 |
| Matsuda index | 6.7 (0.7–34.2) 4.9 |
| **Physical activity measures (Baecke scale), Mean (SD)** | |
| Leisure-time physical activity | 2.9 (0.6) |
| Sports activity | 2.7 (0.9) |
| Work activity | 2.5 (1.0) |
| Total activity | 8.1 (1.5) |

*ALAT* Alanine aminotransferase, *ASAT* Aspartate aminotransferase, *BMI* body mass index, *DZ* dizygotic, *HOMA-IR* Homeostatic model assessment for insulin resistance, *hsCRP* high-sensitivity C-reactive protein, *MZ* monozygotic, *WHR* waist-to-hip ratio.

### Differential methylation analysis

To investigate whether nuclear DNA methylation was associated with mtDNAq, we performed an epigenome-wide association analysis (EWAS). In adipose tissue (n = 153 individuals; 71 MZ pairs, 4 DZ pairs, 3 singletons), we identified one CpG site (*cg19998400*) that was inversely associated with mtDNAq (FDR = 0.002) (Fig. 2a). The CpG is in the enhancer region of *SH3BP4* (SH3 domain binding protein 4) gene that codes for a protein involved in intracellular signaling pathways. Another CpG (*cg17468563*) located in the *DHRS3* (Dehydrogenase/Reductase 3) enhancer region showed marginal association with mtDNAq (FDR = 0.078) (Fig. 2a). Muscle tissue CpG methylation (n = 155 individuals; 74 MZ pairs, 3 DZ pairs, 1 singleton) was not associated with mtDNAq (Fig. 2b). The association profiles of the two tissues did not show similar patterns, as none of the top 100 CpG were found to be in common between adipose and muscle samples. In addition, there was no strong correlation between the

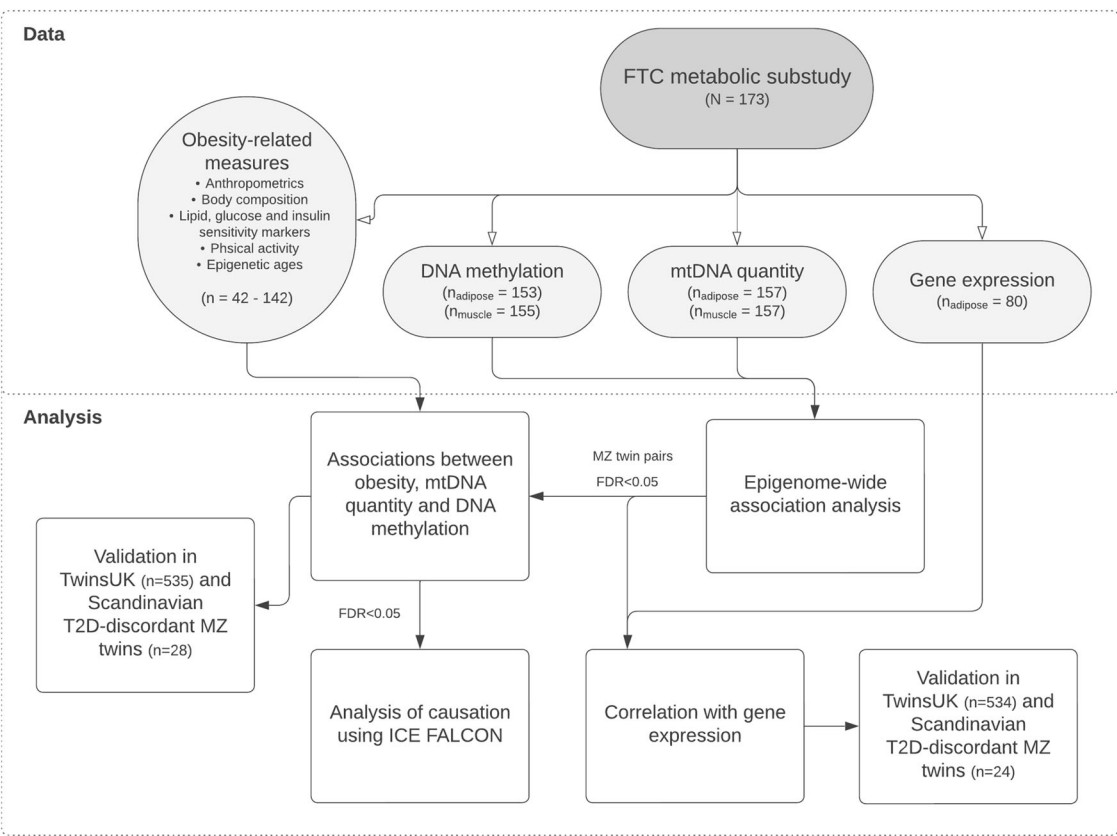

**Fig. 1 | Study flowchart.** FTC Finnish Twin Cohort, mtDNA mitochondrial DNA, MZ monozygotic, T2D type 2 diabetes.

regression effect sizes (Supplementary Fig. 1). The rest of the manuscript focuses explicitly on the findings in adipose tissue, given that we did not detect any statistically significant mtDNAq-linked DNA methylation sites in muscle.

## Causal inference between adipose tissue mitochondrial DNA quantity and identified CpG sites

To investigate the potential evidence for causation underlying the associations between mtDNAq and the two identified CpG sites in adipose tissue, we performed ICE FALCON analysis for the complete MZ twin pairs with data on adipose tissue DNA methylation ($n = 68$ MZ pairs). ICE FALCON is a statistical method that uses familial structure (e.g., twin pairs) to make causal inferences based on changes in regression coefficients[18]. We analyzed each CpG separately, given that we cannot assume them to have identical causal pathways. For instance, the *cg17468563* has been reported to be under the genetic control of multiple loci[19], whereas no meQTLs have been identified for *cg19998400*. Our data was consistent with the hypothesis that mtDNAq is causally linked to *cg19998400* (Table 2). This was suggested when we set mtDNAq as the independent variable and observed a cross-twin cross-trait association (Model 2: $B_{cotwin} = -0.167$, $p = 0.039$) which was attenuated towards null when adjusting for the twin's own mtDNAq. There was no evidence of change to within-individual association between Models 1 and 3. Conversely, when using mtDNAq as the dependent variable, we did not observe any cross-twin cross-trait associations in Models 2 or 3. However, the increased absolute value of the regression coefficient in Model 3 is in line with mtDNAq being causal to methylation at *cg19998400* (Table 2). The change was not statistically significant, which may reflect a low sample size and consequently reduced statistical power. The results from ICE FALCON on *cg17468563* were more ambiguous, with the regression coefficients pointing to either

causation from mtDNAq to methylation at *cg17468563* or the presence of within-individual confounding (Table 2). The results from analysis including also the complete DZ twin pairs ($n = 68$ MZ pairs, 4 DZ pairs) were consistent with the results obtained from MZ twin pairs alone (Supplementary Table 4).

## Causal inference between adipose tissue DNA methylation and gene expression

As DNA methylation is a potential mechanism to regulate gene expression, we investigated whether the DNA methylation at *cg19998400* and *cg17468563* were associated with the expression of their respective genes, *SH3BP4* and *DHRS3* in adipose tissue ($n = 80$ individuals; 38 MZ pairs, 4 singletons). There was a positive correlation between the expression and methylation of *SH3BP4* ($r_{Pearson} = 0.46$, $p < 0.001$) (Fig. 3a), where the methylation explained solely 18% of the variation (marginal R-squared) and with familial confounding 58% (conditional R-squared) of the variation in gene expression (Fig. 3c). For the *DHRS3* expression and methylation, we observed a negative correlation ($r_{Pearson} = -0.66$, $p < 0.001$) (Fig. 3b), with a marginal R-squared of 42% and conditional R-squared 56% (Fig. 3c).

It is important to acknowledge that changes in gene expression patterns can also directly influence DNA methylation levels. Considering this, we conducted an ICE FALCON analysis for the 38 complete MZ pairs to explore the direction of causality between gene expression and DNA methylation. We observed a marginal cross-twin cross-trait association between *SH3BP4* expression and *cg19998400* methylation (Model 2: $B_{cotwin} = -0.112$, $p = 0.078$) (Supplementary Table 5) that attenuated towards null when adjusting for twin's own *cg19998400* levels, whereas the twin's own regression coefficient remained approximately the same. This observation is consistent with *cg19998400* methylation influencing

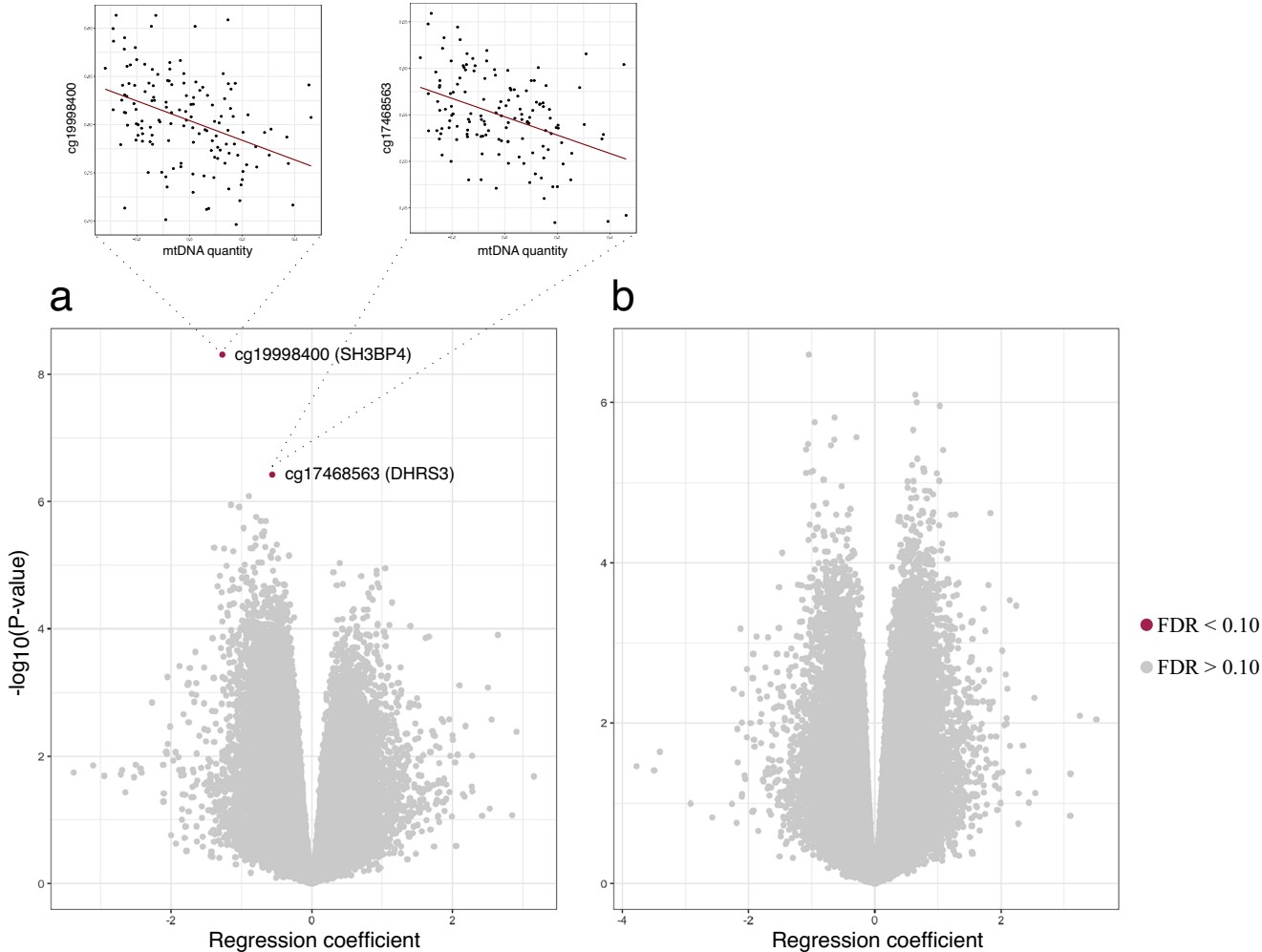

**Fig. 2 | Volcano plots of the epigenome-wide association studies on mtDNA quantity.** Results from (**a**) adipose tissue ($n = 153$ individuals) and (**b**) skeletal muscle ($n = 155$ individuals). Red dots indicate CpGs with FDR < 0.10. The correlations between the highlighted CpGs and mtDNA quantity is plotted above with the red line representing the fitted regression line. The data was analyzed using moderated t-statistics and adjusted for multiple comparisons with false discovery rate (FDR). Source data are provided through Figshare [https://doi.org/10.6084/m9.figshare.26941927.v3]. mtDNA mitochondrial DNA.

the *SH3BP4* expression. On the other hand, the possibility for the presence of within-individual confounding cannot be ruled out. In contrast, with *DHRS3* and *cg17468563*, we did not find any compelling evidence for either causation or familial confounding (Supplementary Table 5).

### Validation of the associations between methylation and expression of *SH3BP4*, and mitochondrial DNA quantity

To validate the association between *cg19998400* methylation and mtDNAq, we analyzed adipose tissue DNA methylation and mtDNAq data from T2D-discordant MZ twin pairs from Scandinavian Twin Registries[20,21] (Supplementary Table 8.). We found a negative association between mtDNAq and *SH3BP4* methylation at *cg19998400* ($p < 0.001$; $n = 14$ MZ pairs), replicating our findings (Supplementary Table 9.). To validate the positive correlation between *cg19998400* methylation and *SH3BP4* expression, we used data from both the Scandinavian T2D-discordant MZ twin cohort ($n = 12$ MZ pairs) and TwinsUK[22] ($n = 79$ MZ pairs, 111 DZ pairs, and 154 singletons) (Supplementary Table 8.). The TwinsUK data showed a significant positive correlation between *cg19998400* methylation and *SH3BP4* expression ($r_{Pearson} = 0.20$, $p < 0.001$). While the Scandinavian T2D-discordant MZ twins also showed a positive effect size, it did not reach statistical significance ($r_{Pearson} = 0.09$, $p = 0.66$).

### Obesity-related variables associated with adipose tissue *SH3BP4* methylation and mitochondrial DNA quantity

Considering the well-established connection between mitochondrial dysfunction and obesity, we were interested to study whether the mtDNAq-associated CpG sites would be also associated with different obesity-related variables. We specifically focused on the DNA methylation of *cg19998400* at the *SH3BP4* locus (herein referred to as *SH3BP4* methylation), as our data suggested a potential causal relationship with mtDNAq, in contrast to *cg17468563* methylation at *DHRS3*. Among a set of 35 obesity-related traits, six showed significant association only with mtDNAq, four with *SH3BP4* methylation, and 14 with both mtDNAq and *SH3BP4* methylation (FDR < 0.05) (Fig. 4 and Supplementary Tables 6, 7).

Three distinct measures of epigenetic age acceleration (EAA), namely Horvath, Hannum, and PhenoAge, displayed a negative association with mtDNAq but not with *SH3BP4* methylation. In addition to EAA, mtDNAq showed negative associations with fat-free mass and adipsin levels, and positive with sports activity. Obesity-related variables that associated exclusively with *SH3BP4* methylation included elevated blood triglyceride levels, hsCRP, fasting glucose levels, and systolic blood pressure. The 14 shared associations between mtDNAq and *SH3BP4* methylation included parameters mainly related to body fat composition, insulin sensitivity, and HDL cholesterol levels. Consistent with existing

**Table 2 | Results from ICE FALCON analysis between mtDNA quantity and the identified CpG sites (n = 68 monozygotic twin pairs) in adipose tissue**

| Formula | Coef* | Model 1 | | | Model 2 | | | Model 3 | | | Change in coefficients | |
|---|---|---|---|---|---|---|---|---|---|---|---|---|
| | | Est | SE | P | Est | SE | P | Est | SE | P | Est | P |
| mtDNAq (Y) cg19998400 (X) | βself | −0.350 | 0.061 | 9.2E-09 | | | | −0.358 | 0.060 | 2.1E-09 | −0.008 | 0.728 |
| | βcotwin | | | | 0.006 | 0.070 | 0.930 | −0.067 | 0.059 | 0.257 | −0.073 | 0.426 |
| cg19998400 (Y) mtDNAq (X) | βself | −0.487 | 0.098 | 7.3E-07 | | | | −0.475 | 0.102 | 3.6E-06 | 0.013 | 0.643 |
| | Bcotwin | | | | −0.167 | 0.081 | 0.039 | −0.055 | 0.072 | 0.444 | 0.112 | 0.351 |
| mtDNAq (Y) cg17468563 (X) | βself | −0.431 | 0.092 | 2.9E-06 | | | | −0.428 | 0.092 | 3.6E-06 | 0.003 | 0.942 |
| | βcotwin | | | | 0.163 | 0.080 | 0.043 | 0.143 | 0.073 | 0.051 | −0.020 | 0.788 |
| cg17468563 (Y) mtDNAq (X) | βself | −0.342 | 0.082 | 2.9E-05 | | | | −0.331 | 0.082 | 5.0E-05 | 0.011 | 0.664 |
| | βcotwin | | | | 0.132 | 0.078 | 0.090 | 0.050 | 0.067 | 0.460 | −0.082 | 0.289 |

*Standardized regression coefficient; β$_{self}$ represents the association between twin's own X and Y variables whereas β$_{cotwin}$ is the cross-twin cross-trait association i.e., the association between twin's own X variable with their co-twin's Y variable. X and Y in brackets indicate whether the variable was used as a predictor or an outcome, respectively. The p-values for models 1–3 were calculated from regression coefficients and standard errors using two-sided z-statistics. The change in coefficients was analyzed using non-parametric bootstrapping. No multiple comparison adjustment was applied. Regression models were adjusted for age, sex, smoking, BMI, and methylation beadchip and row. P-values < 0.05 are bolded.
mtDNAq mitochondrial DNA quantity.

literature, higher mtDNAq were correlated with lower body fat, and higher insulin sensitivity, HDL cholesterol levels, and adiponectin levels.

### Validation of the associations between methylation and expression of *SH3BP4* and body mass index

To validate the observed associations between *SH3BP4* methylation and BMI, we again used data from T2D-discordant MZ twin pairs from Scandinavian Twin Registries and from TwinsUK. With TwinsUK, we replicated the positive association between *SH3BP4* methylation and BMI ($B = 0.061$, $p < 0.001$; $n = 79$ MZ pairs, 112 DZ pairs, and 153 singletons). In line with these findings, the expression of *SH3BP4* was also positively associated with BMI ($B = 0.072$, $p < 0.001$; $n = 131$ MZ pairs, 186 DZ pairs, and 131 singletons). Similar analyses from the Scandinavian T2D-discordant MZ twin cohort ($n = 14$ MZ pairs) produced effect sizes in the same direction but without statistically significant p-values (Supplementary Table 9.). While we initially omitted smoking as a covariate in the Scandinavian cohort analysis due to missing information for ten (35.7%) of the participants, additional analysis with the subsample that included smoking data showed an increased effect size when adjusting for smoking (Supplementary Table 9.). Consistent with our primary and validation findings, Rönn et al. also reported a positive association between BMI and DNA methylation of *cg19998400* at *SH3BP4* in adipose tissue of unrelated female participants[23].

### Causal inference between obesity-related traits, and adipose tissue *SH3BP4* methylation and mitochondrial DNA quantity

To discern the potential causal relationship between the 14 obesity-related variables and both *SH3BP4* methylation and mtDNAq, we employed an ICE FALCON analysis for the complete MZ twin pairs in the cohort (Supplementary Table 3). Our findings indicate that certain variables related to insulin resistance and ectopic fat may exert causal influence on *SH3BP4* methylation, as suggested by significant cross-twin cross-trait association in Model 2 (body fat percentage B$_{cotwin}$ = 0.170, $p = 0.074$; intra-abdominal fat B$_{cotwin}$ = 0.235, $p = 0.031$; HOMA-IR B$_{cotwin}$ = 0.278, $p = 0.002$; Matsuda B$_{cotwin}$ = −0.253, $p = 0.006$; fasting insulin B$_{cotwin}$ = 0.235, $p = 0.004$) that attenuated towards null after conditioning on twin's own corresponding measures (Fig. 5b). The changes in regression coefficients were significant in fat percentage ($p = 0.047$) and marginally significant in Matsuda ($p = 0.062$). Reversing the regression and using *SH3BP4* methylation as a predictor variable also suggested causality from these obesity-related traits to methylation, based on the behavior of the regression coefficients (i.e., the cross-twin cross-trait association increased substantially while the B$_{self}$ remained relatively stable) (Fig. 5c). Results of other obesity-related traits, specifically those measuring body size, liver fat percentage, subcutaneous fat, and HDL cholesterol, were less clear. While the behavior of co-twin's regression coefficients in ICE FALCON may indicate that these variables are a consequence of *SH3BP4* methylation (Fig. 5b, c), the opposing signs between Model 1 and Model 2 coefficients, when using *SH3BP4* methylation as predictor variable X, can also suggest the presence of within-individual confounding (Fig. 5c). Similarly, the ICE FALCON analysis on the association between mtDNAq and obesity-related variables suggested either mtDNAq being causal to most of the obesity-related variables or being subject to unmeasured within-individual confounding (Fig. 5d, e). The ICE FALCON analysis, including both the complete MZ and DZ pairs, replicated the results discussed in this paragraph (Supplementary Fig. 2)

## Discussion

In this study, we identified a significant association between mtDNAq and DNA methylation at *SH3BP4*, which correlated with its gene expression levels. This methylation site was also linked to numerous obesity-related traits, specifically those that measure body fat composition and insulin sensitivity. We further validated the association

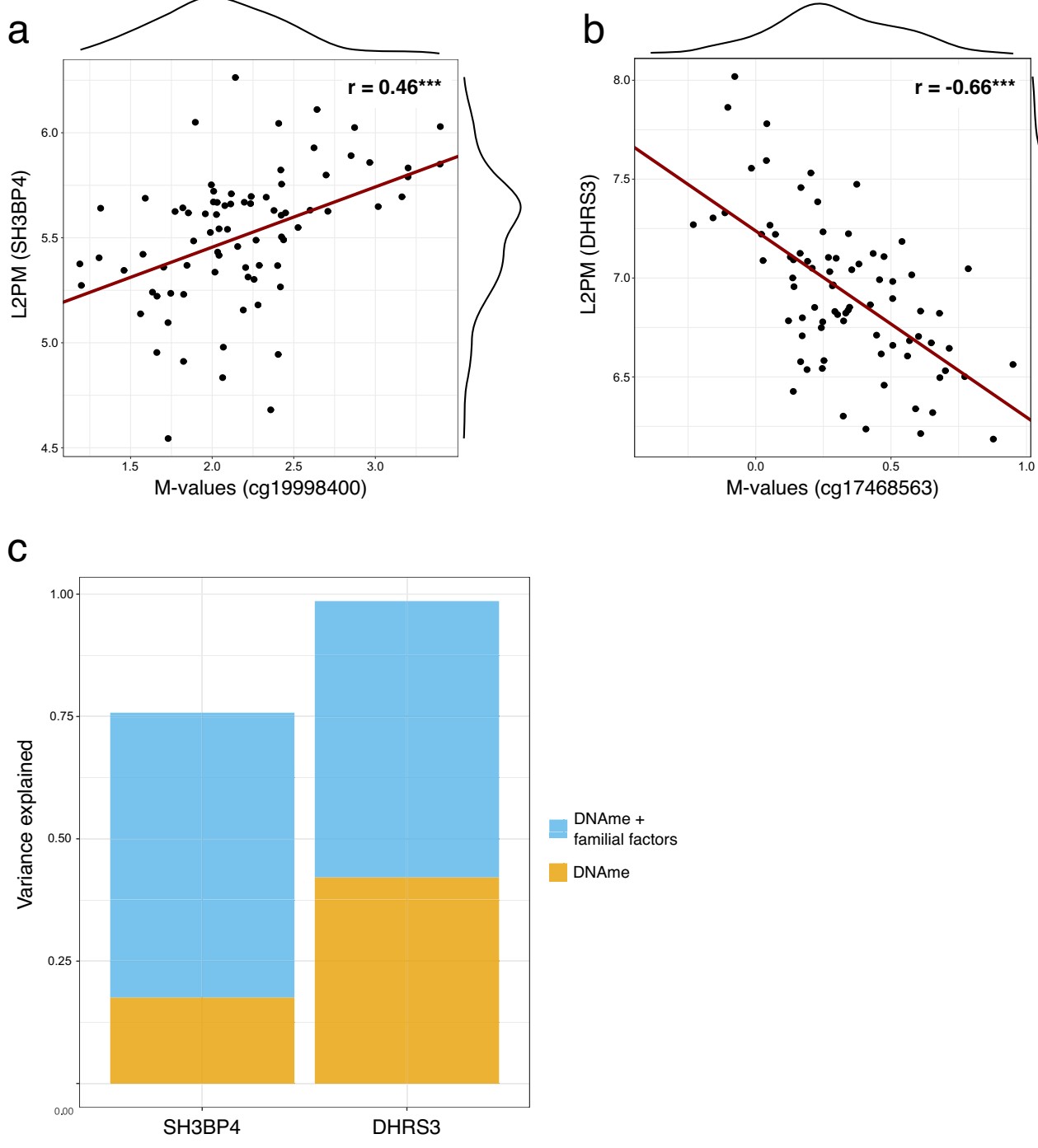

**Fig. 3 | Relationship between DNA methylation and gene expression at two genomic loci in adipose tissue. a** Correlation between methylation at *cg19998400* and *SH3BP4* expression (*n* = 80 individuals), showing the distribution of *cg19998400* on top and *SH3BP4* expression on right. **b** Correlation between methylation at *cg17468563* and *DHRS3* expression (*n* = 80 individuals), showing the distribution of *cg17468563* on top and *DHRS3* expression on right. Red lines in indicate fitted linear regression lines. **c** Variation in *SH3BP4* and *DHRS3* expression explained by methylation at *cg19998400* and *cg17468563* (yellow bars), respectively, and combined with common familial factors (genetic or environmental) (blue bars) (*n* = 80 individuals). Source data are provided in the Source Data file. DNAme DNA methylation, L2PM = log$_2$-counts per million; *r* = Pearson correlation coefficient; *** *p* < 0.001.

between mtDNAq and *SH3BP4* methylation in the Scandinavian T2D-discordant MZ twin cohort, as well as the association between *SH3BP4* methylation, expression, and BMI in TwinsUK. Leveraging our data on MZ twin pairs, we identified potentially causal associations from mtDNAq and obesity-related outcomes to *SH3BP4* methylation in adipose tissue.

The dynamic interplay between mitochondria and the nucleus plays a pivotal role in responding to diverse extracellular signals and metabolic conditions, as suggested by our findings: modifications in mtDNAq may precede changes in *SH3BP4* methylation, which would indicate a retrograde signaling from mitochondria to nuclear DNA methylation. This observation aligns with prior research, although limited, showing the impact of reduced mtDNAq on nuclear DNA methylation in human embryonic kidney cell lines[15]. One of the main hypotheses for retrograde signaling includes the importance of mitochondria in the methionine cycle and in the production of

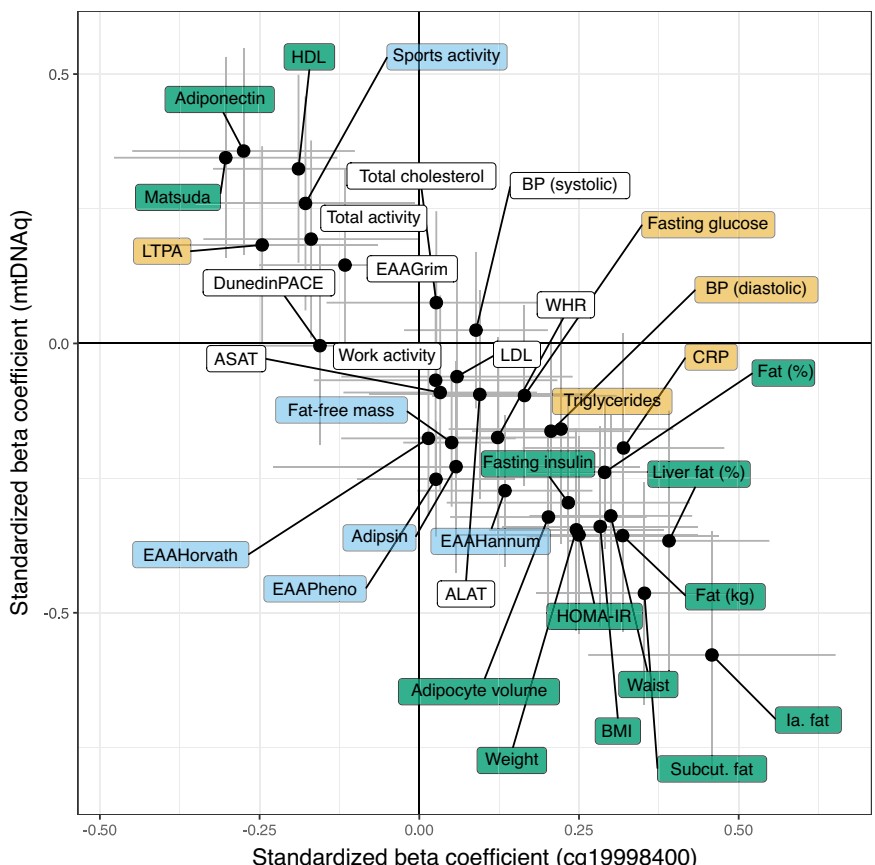

**Fig. 4 | Standardized beta coefficients and their 95% confidence intervals for the associations between obesity-related outcomes and methylation at *cg19998400* (X-axis) or mtDNA quantity (Y-axis) (*n* = 42–142 individuals) in adipose tissue.** Yellow background indicates variables with FDR < 0.05 associated with *cg19998400* methylation only, blue indicates variables associated with mtDNA quantity only, and green indicates variables associated with both. Black dots present estimates for standardized beta coefficients, and gray lines represent 95 % confidence intervals. Source data are provided in the Source Data file. ALAT Alanine aminotransferase, ASAT Aspartate aminotransferase, CRP high-sensitivity C-reactive protein, EAA epigenetic age acceleration, BP blood pressure, HDL high-density, LTPA leisure-time physical activity, mtDNAq mitochondrial DNA quantity, Subcut. subcutaneous.

S-adenosylmethionine. S-adenosylmethionine is a primary methyl donor in cells, which interacts with DNA methyltransferases and therefore can affect DNA methylation[24]. However, the targeted nature of the mitochondria-dependent methylation at specific genomic loci, such as *SH3BP4*, requires further investigation, along with its potential functional implications.

Our study indicated that the DNA methylation of *SH3BP4* and *DHRS3* associated with mtDNAq may be reflected at the gene expression level in adipose tissue. Specifically, methylation of *cg19998400* located at the 5'UTR region of *SH3BP4* correlated positively with *SH3BP4* expression, whereas methylation at the gene body (*cg17468563*) of *DHRS3* displayed a negative correlation, which is contrary to the previously observed general pattern of positive association between gene body methylation and expression[25,26]. However, recent research indicates a far more complex relationship between these two factors, heavily influenced by the underlying genomic context[27–29]. Although our study does not pinpoint the molecular mechanisms underlying the association, we underscore the potential implication of these genes and their methylation status in relation to varying mtDNAq in obesity. Moreover, we demonstrated that DNA methylation of both *SH3BP4* and *DHRS3* accounted for a substantial proportion of the variation in the expression of these genes, alongside shared familial factors within the twin pairs. These factors encompass both genetic elements, such as eQTLs, and environmental factors, like age and lifestyle, which cannot be distinguished apart using MZ twin pairs only. Consequently, future studies with the inclusion of a cohort of DZ twins or other family members could provide insights into the relative significance of genes and environment.

The link we identified between the mtDNAq-associated methylation site in *SH3BP4* and various obesity-related outcomes, particularly those assessing insulin sensitivity and body fat composition, implies a potential role of mtDNAq-induced methylation in the etiology of obesity. *SH3BP4* acts as a negative regulator in many signaling pathways such as mTORC1 (mammalian target of rapamycin complex 1), a key promoter of cell growth and proliferation[30], and has been observed to promote adipogenesis, possibly by regulating mitochondrial functions[31–33]. Previous research has demonstrated that mTOR signaling is compromised in obesity, potentially influenced by factors such as diet quality[34–36] and oxidative stress[37]. Still, the precise contribution of *SH3BP4* methylation and gene expression to cellular function and disease development remains unclear.

Our analysis indicates that changes in DNA methylation at *SH3BP4* may result from alterations in body insulin sensitivity and intra-abdominal fat accumulation. Previous studies conducted on blood have also suggested that methylation may be a consequence rather than a cause of obesity[38,39]. Intriguingly, our findings suggest that only certain obesity-related outcomes, perhaps those more closely tied to metabolic disruptions such as insulin resistance and ectopic fat, are causally associated to *SH3BP4* methylation, while others closely related, such as BMI, are not. This may reflect the high metabolic heterogeneity often observed among people with similar BMI[40]. Nevertheless, the discrepancy warrants further investigation into the

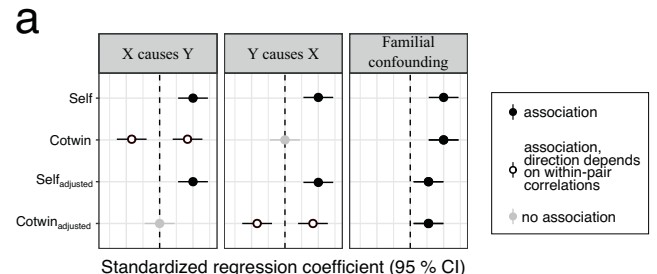

pathways that result in changes in *SH3BP4* methylation in the context of obesity.

The connection between mtDNAq and obesity has been established[13,41,42], yet the exploration of causal pathways has been limited. We identified a dualistic relationship between mtDNAq and obesity-related outcomes, which pointed to either causation from mtDNAq to several obesity-related outcomes or the presence of

unmeasured within-individual confounding. Excessive nutrient intake is a plausible factor influencing this association, as it is known to impair mitochondrial function[43–45] and contribute to the excess body weight. Whether nutrient intake serves as a confounder in the association or initiates a pathway mediated by mitochondria leading to obesity remains uncertain. It is also possible that there is a circular relationship between mtDNAq and obesity. In addition, changes in mtDNAq may

**Fig. 5 | Results from the ICE FALCON analysis between adipose tissue mtDNA quantity, *SH3BP4* methylation and obesity-related outcomes. a** Conceptual figure illustrating the behavior of the ICE FALCON regression coefficients of Models 1–3 in each causal scenario. **b**–**e** Point estimates for the standardized regression coefficients and their 95% confidence intervals for the ICE FALCON analysis of (**b**) obesity-related outcomes regressed against *SH3BP4* methylation, (**c**) *SH3BP4* methylation regressed against obesity-related outcomes, (**d**) obesity-related outcomes regressed against mtDNA quantity and (**e**) mtDNA quantity regressed against obesity-related outcomes. 'Self' represents the association between twin's own *X* and *Y* variables whereas 'Cotwin' is the cross-twin cross-trait association, i.e., the association between twin's own *X* variable with their co-twin's *Y* variable. 'Adjusted' refers to the regression coefficients derived from the ICE FALCON Model 3 that includes both twin's own and their cotwin's *X* variables. The *p*-values were calculated from regression coefficients and standard errors using two-sided z-statistics. No multiple comparison adjustment was applied. Source data with the exact *p*-values are provided as a Source Data file. **b**, **c** N (BMI, Fat %, Fat kg, HDL, Weight) = 68 pairs, N (Fasting insulin) = 64 pairs, N (HOMA-IR, Waist) = 61 pairs, N (Adipocyte vol.) = 57 pairs, N (Matsuda) = 56 pairs, N (Liver fat %) = 41 pairs, N (Adiponectin, Ia. Fat, Subcut. Fat) = 21 pairs **d**-**e** N (BMI, Fat %, Fat kg, Weight) = 71 pairs, N (Fasting insulin) = 67 pairs, N (HOMA-IR, Waist) = 64 pairs, N (Adipocyte vol.) = 60 pairs, N (Matsuda) = 59 pairs, N (Liver fat %) = 42 pairs, N (Adiponectin, Ia. Fat, Subcut. Fat) = 21 pairs HDL high-density lipoprotein, Ia. intra-abdominal, mtDNAq mitochondrial DNA quantity, Subcut. subcutaneous.

manifest only after changes in other mitochondrial parameters, which were not covered in this study, thereby limiting the identification of causal associations between mitochondrial function and obesity and DNA methylation. Despite this, we demonstrate that both mtDNAq as well as specific obesity-related outcomes may be causal to DNA methylation at *SH3BP4*, via shared or independent molecular pathways.

Aging is widely linked with a decline in mitochondrial metabolism, including reduced mtDNAq[46,47]. We revealed an association between mtDNAq and EAA, measured with Hannum, Horvath, and PhenoAge, in adipose tissue, which can indicate that mitochondrial metabolism is one of the key components in driving biological aging. These clocks have been reported to exhibit similar transcriptional signals with one another[48]. While many of the epigenetic clocks were originally developed for whole blood (except for Horvath, that is a multi-tissue clock), it has been shown that some of the clocks, including PhenoAge and Hannum, work fairly robustly in other tissues too[48]. Moreover, PhenoAge shows increased age acceleration in cells with depleted mitochondria[48].

Our study underscores the importance of investigating disease-affected tissues beyond readily available blood samples. Exploring two primary tissues affected by excess weight, we discovered that the associations between DNA methylation and mtDNAq in obesity are not uniform. Only adipose tissue DNA methylation, but not muscle, was found to be associated with mtDNAq, which may reflect the different roles of these two tissues in obesity. For instance, in obesity, alterations in adipose tissue seem to be more profoundly associated with metabolic health than those in muscle[14]. It may be that changes in muscle emerge only after systemic changes, such as insulin resistance, that are followed by lipid accumulation. Furthermore, mtDNAq as a proxy for mitochondrial biogenesis may not be directly comparable between the two tissues.

This study encompasses several strengths. First, the carefully phenotyped twin cohort for a comprehensive range of obesity-related outcomes enables disentangling the most significant molecular phenotypes of obesity to mtDNAq. Second, the inclusion of adipose and skeletal muscle tissues broadens the examination of the impact of excess weight and other obesity-related outcomes across tissues. In addition, using MZ twins, we can use statistical methods such as ICE FALCON to investigate the causality of the observed associations. The ICE FALCON offers a robust approach to explore the causal relationship in the observational data using related individuals, including twins[18,49–51], without the need for genetic instrumental variables. To our knowledge, there is a lack of established genetic variants or polygenic scores to estimate mtDNAq specifically in adipose and muscle tissues. This absence of genetic information on mtDNAq prevented us from using statistical methods such as MR-DoC[52], which depend on genetic data.

However, there are limitations to consider that include the cross-sectional nature of the study, as well as the modest sample size, which substantially limits statistical power. In addition, we lacked precise data on some potential confounding variables, such as dietary habits,

which could potentially affect the observed associations. It is important to note that our cohort comprised mostly participants without severe health conditions, so our findings may not generalize to more serious conditions like metabolic syndrome or T2D. For instance, while the replication analysis between *SH3BP4* methylation and BMI in the Scandinavian MZ T2D-discordant cohort validated the direction of effects, it was statistically nonsignificant. This may be due to low statistical power (only 7 twin pairs were available for the analysis, adjusting for smoking, a known important biological covariate), or cohort-specific differences such as age range and particularly, the higher prevalence of T2D compared to FTC or TwinsUK. Indeed, the link could be validated in a larger Scandinavian cohort, including adipose tissue from 94 unrelated female participants, which also found a positive association between BMI and *SH3BP4* methylation[23].

The characteristics of our dataset serves as both a strength and a limitation: while its distinctive integration of DNA methylation with mtDNAq in adipose and muscle tissues from twins with obesity-related measures represents a comprehensive approach, yet it restricts our ability to extensively validate all our findings in comparable independent datasets, which are very scarce.

Overall, we demonstrate a potential causal link from adipose tissue mitochondrial metabolism to DNA methylation and expression of *SH3BP4*. In addition, this connection holds significance in obesity, where certain outcomes related to insulin sensitivity and intra-abdominal fat were seen to contribute to *SH3BP4* methylation, influenced either by mtDNAq or through alternative pathways. We propose the existence of a complex network interconnecting DNA methylation and mitochondrial metabolism in obesity, contributing to the multi-faceted nature of obesity as a phenotype. Comprehensive examination of their interplay with various metabolic parameters holds promise for advancing our understanding of the intricate metabolic landscape in obesity.

## Methods

The Ethics Committee of the Hospital District of Helsinki and Uusimaa approved the protocols of this FTC substudy data collection (270/13/03/01/2008), and all participants provided their written informed consent. The authors assert that all procedures contributing to this work comply with the ethical standards of the relevant national and institutional committees on human experimentation and with the Declaration of Helsinki.

### Study cohort

The study participants originate from the metabolic substudy from the larger population-based Finnish Twin Cohort (FTC; FinnTwin12[53], FinnTwin16[54], and Older Finnish Twin Cohort[55]). The present substudy was originally designed to study obesity-related metabolism by targeting MZ twin pairs. Twins were invited to participate based on their self-reported weight and height to ensure the presence of twins with varying discordance for BMI. Zygosity was first defined by questionnaires[56], and later confirmed using genotyping, revealing some of the expected MZ pairs to be DZ instead. Participants who

consented to adipose and/or muscle biopsies were included in the study with no other specific inclusion or exclusion criteria beyond the above described. Our data consisted of 173 individuals (81 MZ twin pairs, 5 DZ twin pairs, and 1 MZ singleton without available co-twin data) from this substudy (Supplementary Table 1).

## Clinical data

The selected study participants were deeply phenotyped for obesity-related clinical measures (Table 1) as described more in detail in van der Kolk et al.[14]. Briefly, anthropometric and body composition were measured after overnight fasting. Body mass index (BMI) was calculated from weight and height (kg/m²), measured during on-site visits. In addition, waist circumference was measured and the waist-to-hip ratio (WHR) calculated. Fat mass, fat percentage, and lean mass were quantified using dual-energy X-ray absorptiometry. Intra-abdominal and subcutaneous fat volumes were measured using magnetic resonance imaging (MRI), and liver fat content using magnetic resonance spectroscopy (MRS). Supine blood pressure measurements were also taken (mean of three measurements).

Blood samples were obtained after overnight fasting, and concentrations of plasma glucose, serum insulin, plasma total cholesterol, low-density lipoprotein (LDL), high-density lipoprotein (HDL), triglycerides, high sensitivity C-reactive protein (hsCRP), alanine aminotransferase (ALAT) and aspartate aminotransferase (ASAT) were measured using standard HUSLAB clinical laboratory procedures. Homeostatic model assessment-insulin resistance index (HOMA-IR) and the Matsuda index for insulin sensitivity were calculated from the standard 4-point oral glucose tolerance test (OGTT).

Levels of different forms of physical activity (sport, work, leisure and total) were assessed using the Baecke questionnaire[57,58].

## Sample collection

The samples of this study have been used in previous research[14,59,60]. Briefly, the adipose and muscle tissue samples were collected from subcutaneous abdominal adipose tissue and vastus lateralis muscle, respectively, under local anesthesia (lidocaine). Adipose tissue samples were taken using a surgical technique or needle biopsy, and muscle samples through Bergström needle biopsy. Determination of adipocyte volume (dm³) was determined with the custom algorithm for ImageJ[14] and has been previously described in Lapatto et al.[60]. The tissue samples for DNA/RNA extraction were snap-frozen in liquid nitrogen.

## Mitochondrial DNA quantity

DNA was extracted from adipose and muscle tissue biopsies using AllPrep DNA/RNA/miRNA Universal Kit (Qiagen). The amount of mtDNA was quantified using quantitative PCR (qPCR), targeting for two mitochondrial encoded genes, ND5 (NADH dehydrogenase 5) and CYTB (cytochrome b), and normalized to genomic DNA as measured from APP (amyloid-beta precursor protein) and B2M (beta-2-microglobulin). Details on primer sequences is provided in Supplementary Data 1. Data was processed using the $2^{-\Delta\Delta Ct}$ method with qbase+ software (Biogazelle) to obtain calibrated normalized relative quantities (CNRQ) for the mtDNA quantity. In cases of missing data (due to poor sample quality) for either ND5 or CYTB, we applied a stochastic regression method to input missing values, using the other gene as a reference. Subsequently, we calculated the mean of ND5 and CYTB, which served as the metric for mtDNAq.

## DNA methylation data

High molecular weight DNA was extracted from adipose and muscle biopsies with AllPrep DNA/RNA/miRNA Universal Kit (Qiagen) or QIAmp DNA Mini Kit (Qiagen) and bisulfite converted with an EZ DNA Methylation Kit (ZYMO Research) according to the manufacturers' protocols. DNA methylation was quantified using Illumina Infinium HumanMethylation450K (450K; adipose tissue) or

Illumina HumanMethylationEPIC BeadChip arrays (EPIC; adipose and muscle tissues).

DNA methylation data was preprocessed and normalized with R package meffil[61]. Due to the modest sample size, adipose tissue 450K and EPIC data were preprocessed together, omitting the platform-specific probes from the analysis. After background and bias correction, we excluded bad quality samples with following criteria: (i) Median difference in X and Y chromosome intensities > 3 standard deviations (SDs), (ii) Median methylated vs. unmethylated intensity > 3 SDs, (iii) unreliable control probes, (iv) detection p-value > 0.01 in more than 20% of probes and (v) bead number < 3 in more than 20% of the probes.

We then applied quantile normalization to reduce technical variation by adjusting for methylation sample, slide, and control probe PCs. Number of PCs included was estimated from a scree plot separately for adipose and muscle data (Supplementary Fig. 3). Bad quality probes were removed with following criteria: (i) Detection p-value > 0.01 in more than 20% samples, (ii) bead number < 3 in more than 20% samples, (iii) SNP probes and iv) ambiguous mapping probes[62,63]. After QC, the number of samples and probes were 153/411,585 and 155/765,201 CpG sites for adipose and muscle data, respectively. The data was then beta mixture quantile (BMIQ) normalized to adjust for type2 probe bias. Because a considerable amount of batch effect remained from 450K and EPIC platforms in adipose tissue, we applied ComBat to minimize the effect of a platform (Supplementary Fig. 4). Methylation M-values, calculated as the $\log_2$ ratio between the methylated versus unmethylated probe intensities, were used in the statistical analysis[64].

**Epigenetic age acceleration estimates.** Epigenetic age for each individual was calculated from the preprocessed DNA methylation data. We used the principal component (PC) versions of the original Hannum[65], Horvath[66], GrimAge[67], and PhenoAge[68] clocks, as that has been shown to remove bias caused by technical variation in certain CpGs[69]. Other epigenetic clocks applied were DunedinPACE[70], which measures the pace of aging, as well as muscle-specific epigenetic clock MEAT[71] that was calculated for skeletal muscle tissue only. Epigenetic age acceleration measures, defined as the residuals from regressing an epigenetic age estimate on chronological age, were used in the statistical analyses.

## RNA sequencing data

Adipose tissue RNA sequencing data was available for a subset of the twins (n = 80 individuals). The generation and preprocessing of the RNA-seq data has been described in detail elsewhere[14]. Briefly, RNA was extracted using AllPrep DNA/RNA/miRNA Universal Kit (Qiagen) with DNase I (Qiagen) digestion according to manufacturers' protocol. After the calculation of RNA integrity numbers, the libraries were prepared with Illumina Stranded mRNA preparation and sequenced with the Illumina HiSeq2000 platform. The data was mapped against the human reference genome 38, the quality was calculated, and read counts generated.

## Statistical analysis

**Epigenome-wide association analysis on mitochondrial DNA quantity.** To identify individual CpG sites associated with mtDNAq in adipose and muscle tissue, we performed an EWAS using the R package limma[72] that fits a linear model for each probe and computes moderated Bayes t-statistics. The models were adjusted for known biological and behavioral (age, self-reported sex, smoking status), and technical (beadchip date and row) covariates, as well as cell type proportions. In the absence of reference-based cell type deconvolution methods developed specifically for either adipose or muscle tissues, we applied EpiDISH[73] to adjust for key known cell types: fibroblasts and epithelial cells. In addition, the adipose tissue model was adjusted for the proportion of fat cells derived from EpiDISH. The fraction of immune cells

was omitted from the model due to high correlation with other cell types. The relatedness of twins in a pair was accounted for as a random effect in the model.

**Associations between DNA methylation and gene expression in adipose tissue.** DNA methylation is known to influence gene expression, due to which we investigated whether the identified DNA methylation sites correlated with the expression of their closest genes. We applied Pearson correlation to see to what extent DNA methylation and the expression of their respective genes were related. In addition, using linear mixed effects modeling between gene expression (outcome) and DNA methylation (predictor), and adjusting for the relatedness of the twins, we investigated how much of the variation in gene expression is explained (i) by the changes in DNA methylation solely (marginal R squared) and (ii) together with common familial factors shared within the twin pairs (conditional R squared).

**Associations between obesity-related outcomes, mitochondrial DNA quantity and DNA methylation.** Given that there is a well-established link between mitochondria and obesity, we were interested to explore whether the mtDNAq-associated CpG methylation was further linked to various obesity-related measures. We performed a generalized equation estimation (gee) regression using the R package *geepack*[74] with 'exchangeable' correlation structure to account for within-twin pair similarities. The CpG methylation was used as an outcome variable and each obesity-related outcome separately as a predictor variable. Those obesity-related variables that did not follow normal distribution were log$_{10}$ transformed before the analysis (Supplementary Tables 6, 7). The models were adjusted for age, sex, and smoking status, beadchip date, and row. We determined the associations between the obesity-related outcomes and mtDNAq and chose variables that exhibited statistically significant (FDR < 0.05) links with both mtDNAq and CpG methylation to the causal inference analysis.

**Causal inference.** To assess the potential evidence for a causal relationship or common familial confounding underlying the identified associations between DNA methylation, mtDNAq, and obesity-related outcomes, we applied a statistical method called ICE FALCON (Inference about Causation from Examination of FAmilial CONfounding)[18]. The method is based on regression models of observational data of related individuals, specifically twins, and assesses the changes in twin's own and co-twin's regression coefficient from with and without adjusting the counterparts' predictor variables (Eqs. 1–3 below). As the power of the ICE FALCON depends on the within-pair correlation, we originally restricted the ICE FALCON analysis for the complete MZ pairs, which are expected to be more correlated than DZ pairs. However, we performed additional ICE FALCON analyses that also included the DZ pairs.

$$Y_{self} \sim \alpha + \beta_{self} \times X_{self} \tag{1}$$

$$Y_{self} \sim \alpha + \beta_{cotwin} \times X_{cotwin} \tag{2}$$

$$Y_{self} \sim \alpha + \beta_{self} \times X_{self} + \beta_{cotwin} \times X_{cotwin} \tag{3}$$

Briefly, if the observed cross-twin cross-trait association ($\beta_{cotwin}$) diminishes after adjusting for within-individual association, it suggests a causal effect from the predictor variable to the outcome (Fig. 5a). Conversely, if the $\beta_{cotwin}$ appears only when conditioning on the within-individual association, it indicates a causal relationship from the outcome to the predictor. If both the $\beta_{cotwin}$ and the $\beta_{self}$ decrease after conditioning on each other, it indicates familial confounding (genetic or environmental) between the predictor and outcome. We examined $\beta_{cotwin}$ with and without conditioning on within-individual association

using generalized equation estimation in the R package *geepack*[62] using 'exchangeable' correlation structure for the twin pairs. All analyses were adjusted for age, sex, and smoking. Technical covariates of beadchip date and row were added as covariates to adjust analyse which included DNA methylation data. In addition, the ICE FALCON between mtDNAq and DNA methylation sites were further adjusted for BMI. We then calculated the changes in regression coefficients, for which we estimated standard errors of changes with non-parametric bootstrapping, generating 1000 datasets having the original sample size. The models were then reversed, e.g., using the previous X variable as the Y variable to gain additional evidence on the causal pathway.

**Validation cohorts and statistical analysis.** From the TwinsUK[22] 535 study, participants with available adipose tissue DNA methylation and 765 with gene expression data were selected (Supplementary Table 9) of which 534 had both DNA methylation and gene expression data. Ethical approval for TwinsUK was obtained by the National Research Ethics Service, London-Westminster, the St Thomas' Hospital Research Ethics Committee (EC04/015 and 07/H0802/84). DNA methylation profiles were generated from SAT biopsies using the Illumina Infinium HumanMethylation450K platform[75]. The data was preprocessed for background and dye bias correction, quantile normalized, and probes with detection *p*-value < 0.000001 and bead number < 3 set as missing[76]. The generation of SAT RNA-seq data is described in Christiansen et al. [76]. Briefly, reads were aligned to the hg19 reference genome, and samples with < 10 million reads excluded. Gene counts were inverse-normalized. The associations between *cg19998400* methylation, *SH3BP4* expression, and BMI were performed in R using linear mixed effect modeling, adjusting for age, sex, and three-category smoking status (never, former, and current), technical covariates, and accounting for twin pair similarity as a random effect.

The Scandinavian discordant MZ twin cohort includes fourteen MZ pairs, discordant for T2D, recruited from the Scandinavian twin registries[20,21] (Supplementary Table 8). Ethical approvals were granted by local ethics committees in Fyn (96/253) and Lund (520/2008). Genomic DNA was extracted with a DNeasy Blood and Tissue kit (Qiagen) and subjected to DNA methylation analysis using the Illumina Infinium HumanMethylation450K platform. The methylation data was background corrected, quantile normalized, and probes with detection *p*-value < 0.01 were removed[21]. Total RNA was extracted from frozen SAT biopsies using the miRNeasy kit (Qiagen, Hilden, Germany) and RNA expression analyzed with GeneChip Human Gene 1.0 ST arrays (Affymetrix, Santa Clara, CA), and Robust Multichip Average expression measures were computed[20,21]. Data on SAT mtDNAq was determined with the ViiA 7 system (Applied Biosystems) as the mean value of the quantity of three mitochondrial genes, *ND6*, *RNR2*, and *CYTB*, normalized to the nuclear gene *RNAseP* quantity[21]. The associations between *cg19998400* methylation, *SH3BP4* expression, BMI, and mtDNAq were determined using gee, adjusting for age, sex, and two-category smoking status (yes/no) with 'exchangeable' correlation structure to account for within twin pair similarities. Pearson correlation coefficient was used to determine the correlation between *SH3BP4* methylation and expression.

**Reporting summary**
Further information on research design is available in the Nature Portfolio Reporting Summary linked to this article.

## Data availability
The FTC omics data (DNA methylation and RNA sequencing) is part of the 'Twin Study' and deposited with the Biobank of the Finnish Institute for Health and Welfare [https://thl.fi/en/research-and-development/thl-biobank/for-researchers/sample-collections/twin-

study]. For details on accessing the data, see: https://thl.fi/en/research-and-development/thl-biobank/for-researchers/application-process. The TwinsUK methylation dataset analyzed in the current study is available under ArrayExpress accession number E-MTAB-1866. The expression dataset analyzed in the current study is available under EGA accession number EGAS00001000805. All additional data access requests are overseen by the TwinsUK Resource Executive Committee (TREC). For information on access to these data and how to apply, see: https://twinsuk.ac.uk/researchers/access-data-and-samples/request-access/. The Scandinavian T2D-discordant cohort of MZ twins data (accession number LUDC2020.08.14 [https://www.ludc.lu.se/resources/repository]) are deposited in the Lund University Diabetes Center repository and while summary data and look-ups are available to academic researchers upon request through the repository portal, individual level data are not available due to ethical and legal restrictions related to Swedish Biobanks in Medical Care Act and European GDPR legislation. The EWAS summary statistics on mtDNA quantity in adipose and muscle tissues produced in this study are deposited to Figshare [https://doi.org/10.6084/m9.figshare.26941927.v3]. Source data are provided in this paper.

## Code availability

The codes used in the study to run the EWAS and to perform the ICE FALCON analyses have been deposited to Figshare [https://doi.org/10.6084/m9.figshare.26941927.v3] public repository and are available to download.

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

## Acknowledgements

The authors thank all the study participants. We also acknowledge the computational resources of the Institute for Molecular Medicine Finland (FIMM) Technology Center. Open access was facilitated by Helsinki University Library. S.Lu. was involved in this research while affiliated with the Institute for Molecular Medicine Finland (FIMM) and is currently affiliated with Nightingale Health Plc. This study is supported by the following funds: University of Helsinki, Faculty of Medicine, Doctoral School of Population Health (DOCPOP) (A.He.), an Australian Government Research Training Program (RTP) Scholarship (V.F.C.E.), Academy

of Finland (#328685, #307339, #297908 and #251316, M.O.; S.H.; #335443, #314383, #266286, K.H.P.), Academy of Finland Center of Excellence in Complex Disease Genetics (#352792) (J.K.) and Center of Excellence in Research on Mitochondria, Metabolism and Disease (Fin-MIT) (#272376) (K.H.P.), Sigrid Juselius Foundation (M.O.), Liv o Hälsa society (M.O.), Minerva Foundation (M.O.), Novo Nordisk Foundation (#NNF20OC0060547, #NNF17OC0027232, #NNF10OC1013354, K.H.P.; #NNF23SA0083953, S.H.), National Health and Medical Research Council Investigator Grant (GNT2017373) (S.Li.), Diabetes Research Foundation (S.H., K.H.P.), Paulo Foundation (S.H., K.H.P.), Gyllenberg Foundation (K.H.P.) Finnish Medical Foundation (K.H.P.), University of Helsinki, and Helsinki University Hospital (K.H.P., S.H.), and Government Research Funds (K.H.P.). TwinsUK is funded by the Wellcome Trust, European Community's Seventh Framework Program (FP7/2007-2013), Medical Research Council, Versus Arthritis, European Union Horizon 2020, Chronic Disease Research Foundation (CDRF), Zoe Ltd and the National Institute for Health Research (NIHR)-funded BioResource, Clinical Research Facility and Biomedical Research Center based at Guy's and St Thomas' NHS Foundation Trust in partnership with King's College London. This project also received support from the JPI ERA-HDHL DIMENSION project and UK Biological Sciences Research Council (BBSRC, BB/S020845/1 and BB/T019980/1 to J.T.B). The data generated in the Scandinavian twin cohort were supported by the Swedish Research Council, Region Skåne (ALF), Strategic Research Area Exodiab (Dnr 2009-1039), the Novo Nordisk Foundation, the Swedish Foundation for Strategic Research (Dnr IRC15-0067), the Crafoord Foundation, and the Swedish Diabetes Foundation.

## Author contributions

A. He. and M.O. designed the study. A. He. performed the data analyses, visualized the data, and wrote the first draft of the manuscript. K.H.P., S.H., M.O., and J.Ka. collected and generated the FTC data. A. He. and S.H.T.L. preprocessed the omics data. J.Ku., A.Ha., and J.L. performed the MRI. and P-H.G. the DEXA imaging. P.P. generated the RNA sequencing data. V.F.C.E. and S.Li. developed and assisted on the ICE FALCON analysis. S.V. and J.T.B. replicated the findings in TwinsUK, and A.M., C.L., E.N., and A.V. in the Scandinavian T2D-discordant MZ cohort. M.O. and S.Lu. Supervised the work. All authors critically commented and edited the final version of the manuscript.

## Competing interests

The authors declare no competing interests.

## Additional information

[1]Institute for Molecular Medicine Finland (FIMM), HiLIFE, University of Helsinki, Helsinki, Finland. [2]Minerva Foundation Institute for Medical Research, Helsinki, Finland. [3]Centre for Epidemiology and Biostatistics, Melbourne School of Population and Global Health, University of Melbourne, Melbourne, VIC, Australia. [4]Department of Human Genetics, David Geffen School of Medicine at UCLA, Los Angeles, CA, USA. [5]HUS Medical Imaging Center, Radiology, University of Helsinki and Helsinki University Hospital, Helsinki, Finland. [6]Institute for Clinical Diabetology, German Diabetes Center, Leibniz Center for Diabetes Research, Heinrich Heine University, Düsseldorf, Germany. [7]Public Health Promotion Unit, National Institute for Health and Welfare, Helsinki, Finland. [8]Folkhälsan Institute of Genetics, Folkhälsan Research Center, Helsinki, Finland. [9]Research Program for Clinical and Molecular Metabolism, Faculty of Medicine, University of Helsinki, Helsinki, Finland. [10]Abdominal Center, Nephrology, University of Helsinki and Helsinki University Hospital, Helsinki, Finland. [11]Obesity Research Unit, Research Program for Clinical and Molecular Metabolism, Faculty of Medicine, University of Helsinki, Helsinki, Finland. [12]Department of Internal Medicine, Helsinki University Hospital, Helsinki, Finland. [13]Department of Twin Research and Genetic Epidemiology, King's College London, London, UK. [14]Epigenetics and Diabetes Unit, Department of Clinical Sciences in Malmö, Lund University Diabetes Centre, Scania University Hospital, Malmö, Sweden. [15]Department of Clinical Sciences in Malmö, Lund University Diabetes Centre, Scania University Hospital, Malmö, Sweden. [16]Copenhagen University Hospital, Steno Diabetes Center Copenhagen, Herlev, Denmark. [17]Department of Endocrinology, Skåne University Hospital, Malmö, Sweden. [18]Bioinformatics Interdepartmental Program, UCLA, Los Angeles, CA, USA. [19]Institute for Precision Health, David Geffen School of Medicine at UCLA, Los Angeles, CA, USA. [20]HealthyWeightHub, Endocrinology, Abdominal Center, Helsinki University Central Hospital and University of Helsinki, Helsinki, Finland. [21]Precision Medicine, School of Clinical Sciences at Monash Health, Monash University, Clayton, Victoria, Australia. ✉e-mail: aino.heikkinen@helsinki.fi; miina.ollikainen@helsinki.fi

