## [Transparent Peer Review file · Nature Communications]

Twin pair analysis uncovers links between DNA methylation, mitochondrial DNA quantity and obesity

Corresponding Author: Ms Aino Heikkinen

Version 0:

Reviewer comments:

Reviewer #1

(Remarks to the Author)

The authors conducted a twin study to uncover the links between DNA methylation, mitochondrial DNA quantity and obesity. This topic is interesting. My comments are as follows:

1. As the authors described, "A total of 173 individuals participated in the study, comprising 81 MZ twin pairs and 5 DZ twin pairs". It seemed that a total of 172 individuals were included. Could the authors explain this more clearly? How did the authors deal with the singleton in statistical analysis?
2. Both MZ twin pairs and DZ twin pairs were included in the EWAS analysis. But the following analyses were performed in MZ twin pairs. Why did the authors change the participants? Have the authors considered the effect of genetic or environmental factors on the DNA methylation variation identified in EWAS? I suggest performing a sensitivity analysis where just MZ twin samples are used.
3. Could both the MZ and DZ twin pairs be included in Causal inference analysis using ICE FALCON method?
4. What were the inclusion and exclusion criteria for selecting participants in this study?
5. In EWAS analysis, did the authors consider other covariates, such as drinking status and dietary status? What was the criterion of genome-wide statistically significant? Please add this in the text.
6. Did the author perform a normality test on the data?

Reviewer #2

(Remarks to the Author)

In this manuscript authors analyze data from the subcohort of the Finnish Twin cohort (n=173 individuals, with a focus on epigenetic associations in adipose and muscle tissues related to mitochondrial DNA quantity (mtDNAq) and metabolic traits. The authors conclusion relies mainly on adipose tissue, where only one specific CpG site showed statistically significant associations with mtDNAq. However, no significant associations were found in muscle tissue. The focus on adipose tissue methylation without similar findings in muscle tissue raises questions about tissue-specific effects. The authors claim to identify associations between DNA methylation, gene expression, and metabolic traits, however the underlying biological mechanisms details are lacking, highlighting the need for further investigations to elucidate molecular mechanisms underlying the observed associations. The study has very limited novel insights. Major limitations include sample size constraints, lack of the significant tissue-specific effects, challenges in establishing causality, and the need for further elucidation and validation of underlying biological mechanisms.

Reviewer #3

(Remarks to the Author)

In this paper, "Twin pair analysis uncovers novel links between DNA methylation, mitochondrial DNA quantity and obesity," Heikkinen et al. used data from the Finnish Twin Cohort (n=173; 86 full twin pairs) that includes comprehensive measurements of obesity-related outcomes, mitochondrial DNA quantity (mtDNAq) and nuclear DNA methylation levels in adipose and muscle tissue. They identified one locus at SH3BP4 43 (cg19998400) significantly associated with mtDNAq in adipose tissue (FDR<0.05).

While the paper is intriguing, and a good use of the data from the Cohort, there are a few questions that remain:

- 1) It is confusing in general, why the author only used the MZ twin pairs. They mention they have also included DZ twin pairs, but nowhere in the paper I can find any mention of them using this patient cohort. In fact, it would have been advantageous to use the DZ twin pairs to compare the findings between MZ and DZ. If the authors can provide this analysis with this paper it will greatly strengthen the paper.
- 2) Abstract: lines 49 - 51: the sentence needs to be reworded.
- 3) Results. pg. 3 line 106: fine DZ as dizygotic, since it is the first time discussing.
- 4) Results: pg. 3 line 109: remove "with"
- 5) Results: pg 4, lines 118 - 130: It is rather difficult to assess with the volcano plots how the mtDNAq is associated with the methylation data. There seems to be only 2 CpG sites that significantly regulated. But the authors still can create visuals to determine correlation clearly. For example, they create correlation plots instead...
- 6) Results: pg. 5 line 143: remove "in the"
- 7) Results: pg. 7 lines 191 - 193: The authors need to provide the exact name for each of these that appear in Figure 4. It will provide better clarity and more efficiently to find these what is stated here in Figure 4.
- 8) There is no reference for Figure 5a in the paper.
- 9) Methods: pg. 14 Mitochondrial DNA quantity: it doesn't make sense why the authors chose APP and B2M a factor to normalize to for the mtDNA. Please justify this. Are the authors assuming there will be no APP or B2M in the muscles and adipose tissue?? This factor might change.
- 10) Did they check for technical variation of the methylation measures? These can be quite noisy.
- 11) The authors make note of the data availability, but where is the code used for analysis of these data? There should be a GitHub or related repository (e.g. Zenodo) for these methods.

Reviewer #4

(Remarks to the Author)

The authors of this paper conducted a study to determine how mtDNA quantity impacts obesity outcomes in relation to key epigenetic changes in mitochondrial genes. They utilized a cohort of twins to examine differences within individuals compared to their twins. SH3BP4 was identified as the top gene associated with obesity-related traits, showing significant correlations with both the methylated region and mtDNA quantity

There are some overall major concerns with this study:

1. Inclusion of DZ Twin Pairs: It is unclear why the authors only used monozygotic (MZ) twin pairs when they mention including dizygotic (DZ) twin pairs as well. The paper does not provide any analysis involving DZ twin pairs, which would be advantageous for comparing findings between MZ and DZ twins. Including this analysis would greatly strengthen the paper.
2. Definition of "Familial Factors": The term "familial factors" is mentioned in Figure 2 and other sections, but it is not clearly defined. The authors need to specify what they mean by this term, what factors are included, and how these factors were used in the analysis. This lack of clarity makes the discussion confusing. I could maybe assume this is what is described in table 1.
3. Impact of Type 2 Diabetes: The cohort includes 22 participants with type 2 diabetes, which can significantly impact the results. It is unclear whether both twins in each pair had diabetes or only one. Analyzing the data by splitting the participants into diabetic and non-diabetic groups would provide important insights and likely yield different results for these populations.
4. Volcano Plots and CpG Sites: On page 4, lines 118-130, the volcano plots do not clearly show how mtDNA quantity is associated with methylation data. Only two CpG sites are significantly regulated, which is unexpectedly low. The authors should provide correlation plots to better visualize these associations and explain why so few CpG sites are regulated. Is this due to poor statistical power or other factors?
5. Introduction of ICE FALCON: When introducing ICE FALCON in the results, the authors should provide a more detailed explanation of this model. A few sentences describing the model would help readers better understand its application and significance.
6. Normalization Factors for mtDNA: On page 14, the authors chose APP and B2M as normalization factors for mtDNA quantity. This choice needs justification, as it is unclear why these factors were selected and whether they are present in muscle and adipose tissue. The authors should explain their assumptions and the rationale behind this choice.

Additionally, some minor points need to be addressed:

1. Abstract: Lines 49-51 need rewording for clarity.
2. Results: On page 3, line 106, define DZ as dizygotic since it is the first mention.
3. Results: On page 3, line 109, remove "with."
4. Results: On page 5, line 143, remove "in the."
5. Results: On page 7, lines 191-193, provide the exact names for each item in Figure 4 for clarity.
6. Figure Reference: There is no reference to Figure 5a in the paper; this needs to be included.

Version 1:

Reviewer comments:

Reviewer #1

(Remarks to the Author)

The authors have responded the comments. I still have several comments as follows:

In this study, all twins were treated as individual subjects in EWAS analysis, whereas the ICE FALCON analysis was performed using MZ and DZ twins. The twin-specific characteristics were only considered in the ICE FALCON analysis, not in the EWAS. I recommend the authors conduct the EWAS analysis using twin models to better control for genetic and environmental influences.

(Remarks on code availability)

Reviewer #2

(Remarks to the Author)

After reviewing the authors' responses to the initial comments, I acknowledge the efforts made to address several key points raised. The authors emphasize the novelty of their work, particularly in linking DNA methylation, mitochondrial metabolism, and obesity across two relevant tissues—adipose and muscle. As such using multi-tissue approach and the potential modeling of causal pathways between associations is a valuable contribution. However, some of the initial concerns remain only partially addressed. While the authors highlight their findings in adipose tissue as evidence of tissue-specificity, however such an unexpectedly low association (one CpG only) and absence of significant results in muscle tissue continues to raise questions about the robustness of these associations. Specifically when there is not validation strategy in place with respect to identified gene “SH3BP4”. Additionally, although the authors have mentioned modeling causal pathways, the underlying biological mechanisms still require further elucidation, which they acknowledge is a future research direction. The sample size remains a limitation, potentially affecting the power to detect broader tissue-specific effects and mechanistic insights. In conclusion, while the authors have made important clarifications and defended the novelty of their study, further work (e.g. validation / functional analysis with respect to SH3BP4 gene in the context of obesity) is needed to address the biological mechanisms, which would strengthen the overall impact of the findings.

(Remarks on code availability)

Reviewer #3

(Remarks to the Author)

The authors have addressed all my points, and the only remaining issues I see are:

- 1) ideally the links to genes with associations like SH3BP4 should have more links to other published work, or even if possible a replication cohort. Were they able to show any evidence of this from other groups?
- 2) All the code, methods, and scripts need to be clearly organized and public. The link they provided, <https://doi.org/10.6084/m9.figshare.26941927.v1>, only has the R session and two text files, but where is the raw data? Perhaps I missed it, but this should also be present and links to the code page.

(Remarks on code availability)

Reviewer #4

(Remarks to the Author)

The authors have addressed all if my concerns and I believe should be ready for publication.

(Remarks on code availability)

They have provided the code publicly.

Version 2:

Reviewer comments:

Reviewer #1

(Remarks to the Author)

The authors have addressed my concern and I believe that this manuscript can be ready for publication.

(Remarks on code availability)

Reviewer #2

(Remarks to the Author)

The authors have responded well and in the revised manuscript they have satisfactorily addressed my comments by replicating the findings in the additional validation datasets.

(Remarks on code availability)

NA

We are grateful to all the reviewers for their insightful and observant comments. In response to the feedback, we have made substantial edits and clarifications to the manuscript, which are highlighted in yellow. In addition, here we provide detailed, point-by-point responses to each of the reviewers' questions and concerns.

REVIEWER COMMENTS

Reviewer #1 (Remarks to the Author):

The authors conducted a twin study to uncover the links between DNA methylation, mitochondrial DNA quantity and obesity. This topic is interesting.

We are pleased to hear that the reviewer found the topic interesting and are thankful for their efforts in reviewing the paper. Below, we have addressed the concerns raised.

My comments are as follows:

1. As the authors described, "A total of 173 individuals participated in the study, comprising 81 MZ twin pairs, and 5 DZ twin pairs". It seemed that a total of 172 individuals were included. Could the authors explain this more clearly? How did the authors deal with the singleton in statistical analysis?

We agree that the description of the dataset used in the study was not adequate and the numbers could be confusing. Therefore, we have revised the text to clarify the inclusion of singletons (those twins whose co-twin's data was not available for the study) and added a follow-up sentence specifying the exact data used in the subsequent analyses (see pg. 3-4, lines 106-110).

Regarding the use of singletons, our EWAS analysis included three singletons for the adipose tissue analysis and one single singleton for the muscle analysis. In the EWAS analysis, all twins were treated as individuals through introducing a random effect for each pair, as now clarified in pg. 17, line 468. The EWAS was not performed using within-pair models. The singletons were later excluded from the ICE FALCON analysis, which requires data from both co-twins.

2. Both MZ twin pairs and DZ twin pairs were included in the EWAS analysis. But the following analyses were performed in MZ twin pairs. Why did the authors change the participants? Have the authors considered the effect of genetic or environmental factors on the DNA methylation variation identified in EWAS? I suggest performing a sensitivity analysis where just MZ twin samples are used.

In the EWAS, our goal was to maximize statistical power by including as large sample size as possible. Therefore, DZ twins were also included in the analysis. While we agree that comparing models using both MZ and DZ pairs to models using only MZ pairs can provide insights into the impact of genes versus environment on the observed associations, such a comparison cannot be reliably made with the limited number of only five DZ pairs in our study.

We recognize the advantage of performing a sensitivity EWAS using only MZ pairs to confirm that the findings are not driven by the few DZ pairs in the data. As

expected, the significant CpG in the SH3BP4 gene (cg19998400) in the adipose tissue remained significantly associated with mtDNA quantity in the MZ-only model and exhibited very similar regression coefficient (see table below).

EWAS model	Effect size	P-value	FDR
Both MZ and DZ	-1.271	4.9E-09	0.002
MZ-twins only	-1.246	8.7E-08	0.018

We chose to use only MZ pairs in ICE FALCON analyses in attempt to maximize the chance of observing true causal effects. To conclude that ICE FALCON results are consistent with causation from predictor to outcome, we are required to observe a cross-twin cross-trait association. The magnitude of a cross-twin cross-trait association due to causation is influenced by the size of the correlation within-pairs for both the predictor and outcome variables. We hypothesized that using only MZ pairs would produce stronger within-pair correlations than if both MZ and DZ pairs were used, with minimal risk of losing power by excluding only four DZ pairs.

3. Could both the MZ and DZ twin pairs be included in Causal inference analysis using ICE FALCON method?

Yes, both MZ and DZ twins can be included in the ICE FALCON analysis, even though we initially restricted our analysis to MZ twins only, as described above. However, we have now expanded the analyses to include DZ pairs, thereby increasing the sample size by eight individuals (four full pairs). The results are presented in Supplementary Figure 2 and Supplementary Table 4) and indicate that the interpretation of the models remain the same. There are minor increases in the strength of some cross-twin cross-trait associations, especially between mtDNA quantity, and methylation at cg19998400, BMI and waist circumference, which may reflect increased statistical power. Therefore, we conclude that the ICE FALCON results do not show major changes whether we use only MZ twins, or both MZ and DZ twins in the current study.

4. What were the inclusion and exclusion criteria for selecting participants in this study?

We have added sentences in the manuscript (pg. 13, lines 363-366) explaining that participants were screened based on their self-reported BMI and zygosity, with the aim of recruiting MZ twin with varying degrees of within-pair discordance for BMI. We also added a statement that all participants who consented to adipose and/or muscle biopsies, and thus their DNA could be used to generate DNA methylation data from these tissues, were included in the study (pg. 13, lines 366-368). There were no specific additional inclusion or exclusion.

5. In EWAS analysis, did the authors consider other covariates, such as drinking status and dietary status? What was the criterion of genome-wide statistically significant? Please add this in the text.

We considered the inclusion of the most well-known potential confounding factors (age, self-reported sex and smoking) in the EWAS models. We did not include drinking or dietary status as covariates in the analysis, partly because of the small sample size and the risk of overfitting, and partly because of the lack of precise enough data (i.e. on dietary status). In addition, our cohort consists mainly of light drinkers (on average 0.28 alcoholic drinks / day), nor do we believe that light to moderate alcohol consumption has much of an effect on the relationship between mitochondrial DNA quantity and DNA methylation. We argue that including alcohol consumption in the models would not change the results drastically. It would however further reduce the sample size and statistical power as 23 twin individuals were missing data on alcohol consumption at the time of sampling. Nevertheless, for individuals with self-reported data on alcohol consumption, we performed an additional EWAS, adjusting for the alcohol consumption, and observed very similar effect sizes for both SH3BP4 (-1.26 vs. -1.27) and DHRS3 methylation (-0.52 vs. -0.56) in the alcohol-adjusted vs. the original model.

Regarding the inclusion of dietary status as a covariate, we acknowledge that nutrient status can impact metabolism, as mentioned in pg. 10, lines 277-279. However, the data we have on diet for a subset of participants is inconsistent across the study participants, coming from either food frequency questionnaires or three-day food diaries. Consequently, incorporating this data would likely add no or only little valuable information, and could increase the noise in the worst case. We note this as a limitation in our discussion (pg. 12, lines 342-344).

We implemented the Benjamini-Hochberg $FDR < 0.05$ as the genome-wide statistical significance level. Sites passing this significance level were included in the downstream ICE FALCON analysis. However, due to the modest statistical power and the explanatory nature of the study, we decided to also highlight CpGs with $FDR < 0.10$ in the text and discuss their known functions without including them in the downstream analysis.

6. Did the author perform a normality test on the data?

We assessed the normality of our data (obesity-related variables and mitochondrial DNA quantity) graphically by using QQ-plots and histograms. When we observed skewness or other signs of non-normality, we applied \log_{10} transformation to the data and reassessed normality using the same graphical methods. At the end of this response letter, we have attached plots that show the data distribution by histograms and QQ-plots, as well as \log_{10} transformed distribution for those variables that were transformed.

While statistical tests such as the Shapiro-Wilk can be used to evaluate normality, they are often overly strict. Furthermore, the generalized estimating equations (GEE) method we used in the study is robust with non-normally distributed data (Liang & Zeger, 1986). Thus, we believe that the data distribution does not have a significant impact on our results.

Reviewer #2 (Remarks to the Author):

In this manuscript authors analyze data from the subcohort of the Finnish Twin cohort (n=173 individuals, with a focus on epigenetic associations in adipose and muscle tissues related to mitochondrial DNA quantity (mtDNAq) and metabolic traits. The authors conclusion relies mainly on adipose tissue, where only one specific CpG site showed statistically significant associations with mtDNAq. However, no significant associations were found in muscle tissue. The focus on adipose tissue methylation without similar findings in muscle tissue raises questions about tissue-specific effects. The authors claim to identify associations between DNA methylation, gene expression, and metabolic traits, however the underlying biological mechanisms details are lacking, highlighting the need for further investigations to elucidate molecular mechanisms underlying the observed associations. The study has very limited novel insights. Major limitations include sample size constraints, lack of the significant tissue-specific effects, challenges in establishing causality, and the need for further elucidation and validation of underlying biological mechanisms.

We thank the reviewer for highlighting important aspects of our work. While we agree with some of the limitations noted and have discussed them openly in the manuscript, we argue that this research has significant novelty value. To our knowledge, this is a pioneering study that links DNA methylation, mitochondrial metabolism, and obesity in two main obesity-affected tissues, adipose and muscle. Moreover, we have successfully modeled potential causal pathways between the observed associations. Therefore, we believe that this study has a significant impact on the field of human biology and medicine, paving the way for further research to validate the causal pathways, and to provide molecular level mechanisms underlying the findings we report here – research that would not be possible without our study guiding the way.

Additionally, we do not fully agree with the notion of a “lack of tissue-specific effects”. We conducted a comparison between adipose and muscle tissue, and found significant associations only in adipose tissue, which we argue constitutes a tissue-specific effect. Additionally, we compared the effect sizes of all the CpGs between the two tissues (Supplementary Figure 1) and observed almost no correlation between them. This lack of correlation further supports the tissue-specific role of DNA methylation on mitochondrial DNA quantity.

Reviewer #3 (Remarks to the Author):

In this paper, “Twin pair analysis uncovers novel links between DNA methylation, mitochondrial DNA quantity and obesity, “ Heikkinen et al. used data from the Finnish Twin Cohort (n=173; 86 full twin pairs) that includes comprehensive measurements of obesity-related outcomes, mitochondrial DNA quantity (mtDNAq) and nuclear DNA methylation levels in adipose and muscle tissue. They identified one locus at SH3BP4 43 (cg19998400) significantly associated with mtDNAq in adipose tissue (FDR<0.05).

While the paper is intriguing, and a good use of the data from the Cohort, there are a few questions that remain:

We appreciate that the reviewer found our paper intriguing and acknowledges the benefits of our study cohorts. Here we have addressed the remaining questions

below.

1) It is confusing in general, why the author only used the MZ twin pairs. They mention they have also included DZ twin pairs, but nowhere in the paper I can find any mention of them using this patient cohort. In fact, it would have been advantageous to use the DZ twin pairs to compare the findings between MZ and DZ. If the authors can provide this analysis with this paper, it will greatly strengthen the paper.

We recognize that the use of DZ was not clearly stated in the manuscript and have now specified the number of MZ and/or DZ pairs used in each analysis (pg. 3-4, lines 107-110; pg. 4, lines 124 & 129-130; pg. 6, lines 163-164 & lines 174).

In the EWAS analysis, we indeed included DZ pairs, but later excluded them from the subsequent ICE FALCON analysis (see Reviewer #1, Question 2). We fully agree that comparing results between MZ and DZ twins would provide further insights into the role of genetics and environment. However, with only five full DZ pairs, such comparisons cannot be reliably made.

2) Abstract: lines 49 - 51: the sentence needs to be reworded.

We have reworded it now.

3) Results. pg. 3 line106: fine DZ as dizygotic, since it is the first time discussing.

This has now been corrected in the manuscript (pg. 3, line 105).

4) Results: pg. 3 line 109: remove "with"

We have decided to keep the phrase as it is because we prefer to keep the people-first style of language. We use 'sample with overweight' rather than 'overweight sample' to reduce the stigma associated with obesity by not defining people by their body weight.

5) Results: pg 4, lines 118 - 130: It is rather difficult to assess with the volcano plots how the mtDNAq is associated with the methylation data. There seems to be only 2 CpG sites that significantly regulated. But the authors still can create visuals to determine correlation clearly. For example, they create correlation plots instead...

We identified two differentially methylated CpGs in adipose tissue and none in muscle, as illustrated in the volcano plot. Given our sample size and the multiple correction procedure, we did not expect to detect many more significant CpGs without increasing the risk of Type I errors (false positive). A previous study discovered 34 CpGs associated with mtDNA quantity in blood using a sample of 2,507 study participants (Castellani et al. 2020), demonstrating that even with a larger sample size, the number of significant CpGs is limited.

We believe that the volcano plots effectively communicate our main message by showing the number and effect sizes of CpGs associated with mtDNA quantity in both tissues. We acknowledge that the correlation between the two significant CpGs

and mtDNA quantity is not visualized effectively through such plots. Therefore, we have added line plots to Figure 2 to better illustrate this relationship (pg. 30).

6) Results: pg. 5 line 143: remove "in the"

We have now clarified the sentence based on the comment (pg. 5 line 148).

7) Results: pg. 7 lines 191 - 193: The authors need to provide the exact name for each of these that appear in Figure 4. It will provide better clarity and more efficiently to find these what is stated here in Figure 4.

We have listed the exact names of the variables that appear on Figure 4 as footnotes. We hope this improves the clarity as requested by the reviewer.

8) There is no reference for Figure 5a in the paper.

Thank you for pointing this out. We have now added references for Figure 5a in two different occasions (pg. 8, line 210; pg. 19, line 513).

9) Methods: pg. 14 Mitochondrial DNA quantity: it doesn't make sense why the authors chose APP and B2M a factor to normalize to for the mtDNA. Please justify this. Are the authors assuming there will be no APP or B2M in the muscles and adipose tissue?? This factor might change.

When generating data on relative mitochondrial DNA amount, we normalize the measured mitochondrial DNA amount (ND5 and CYTB) to the measured nuclear DNA (APP and B2M) amount, as we mentioned on pg. 14 lines 403-406. Unlike in gene expression quantification, we measure the DNA copies of these genes rather than their expressed RNA. Therefore, APP and B2M, two nuclear genes present in any human cell with DNA, serve our purpose well.

10) Did they check for technical variation of the methylation measures? These can be quite noisy.

Methylation data can be influenced by technical variation such as the platform used, run date, and run slide. To mitigate these effects, we implemented a rigorous pre-processing pipeline. Specifically, we used functional normalization, which corrects methylation values based on a selected number of principal components calculated from the control probes that are present in Illumina arrays (Supplementary Figure 3) and adjusts for the run slide variable. Additionally, we adjusted for additional technical covariates in our EWAS (row and run date). These steps ensure that our results are robust and minimally affected by technical variation.

11) The authors make note of the data availability, but where is the code used for analysis of these data? There should be a GitHub or related repository (e.g. Zenodo) for these methods.

The EWAS and ICE FALCON codes are now available in Figshare (<https://doi.org/10.6084/m9.figshare.26941927.v1>).

Reviewer #4 (Remarks to the Author):

The authors of this paper conducted a study to determine how mtDNA quantity impacts obesity outcomes in relation to key epigenetic changes in mitochondrial genes. They utilized a cohort of twins to examine differences within individuals compared to their twins. SH3BP4 was identified as the top gene associated with obesity-related traits, showing significant correlations with both the methylated region and mtDNA quantity.

There are some overall major concerns with this study:

We are grateful for the insightful comments from the reviewer. We have addressed the concerns below.

1. Inclusion of DZ Twin Pairs: It is unclear why the authors only used monozygotic (MZ) twin pairs when they mention including dizygotic (DZ) twin pairs as well. The paper does not provide any analysis involving DZ twin pairs, which would be advantageous for comparing findings between MZ and DZ twins. Including this analysis would greatly strengthen the paper.

Please refer to Reviewer #3 Question 1 for a detailed response to this question.

2. Definition of "Familial Factors": The term "familial factors" is mentioned in Figure 2 and other sections, but it is not clearly defined. The authors need to specify what they mean by this term, what factors are included, and how these factors were used in the analysis. This lack of clarity makes the discussion confusing. I could maybe assume this is what is described in table 1.

We appreciate the reviewer's observation regarding the need for a clearer definition of "familial factors" in our paper. As we mention on pg. 10, lines 265-267, we use familial factors as a term to describe any known or unknown attribute that is shared by the co-twins, including e.g. genetics, early environmental influences, and socioeconomic status. However, we are not able to disentangle which part of these familial factors drive the association, but only discuss their potential role in explaining it.

We assume that the reviewer refers to Figure 3 in which we present the variation explained in gene expression by DNA methylation and familial factors. By applying mixed effects models, such as GEE, we are able to decompose the variance explained by both fixed and random effects. In our study as shown in Figure 3, the variation in gene expression was attributed to fixed effects (DNA methylation) and random effects (twin pairs). Therefore, the variation explained by random effects corresponds to the variation in gene expression due to familial factors, i.e. any shared factors within twin pairs.

3. Impact of Type 2 Diabetes: The cohort includes 22 participants with type 2 diabetes, which can significantly impact the results. It is unclear whether both twins in each pair had diabetes or only one. Analyzing the data by splitting the participants

into diabetic and non-diabetic groups would provide important insights and likely yield different results for these populations.

Among the 22 diabetic individuals in the study, there were 6 pairs in which both co-twins had diabetes (12 individuals in total), and 10 individuals who had a non-diabetic co-twin. We included the twins with diabetes in the study because 1) diabetes represents the extreme end of the obesity-related phenotypes we are investigating, and 2) the individuals with diabetes were diagnosed during the study visit, meaning they were not aware that they had diabetes nor were they on diabetes medication at the time of sampling.

We agree with the reviewer that comparing diabetic to non-diabetic participants could likely provide novel insights into the pathology of obesity and diabetes, but due to the modest sample size, we could not perform a reliable, large-scale EWAS on this subset with diabetes alone. Instead, we examined whether including twins with diabetes would influence our findings in adipose tissue by 1) performing the EWAS while adjusting for diabetes status, and 2) analyzing the identified CpG site (cg19998400) at the SH3BP4 separately in diabetic versus non-diabetic participants.

The EWAS adjusted for diabetes status replicated our findings on SH3BP4 methylation (see table below). In addition, nine out of the top ten CpGs, including cg17468563 at DHRS3, were shared with the original model used in the manuscript which indicates that the reported findings are not solely driven by the individuals with diabetes.

EWAS model	Effect size	P-value	FDR
Original	-1.271	4.9E-09	0.002
T2D-adjusted	-1.250	7.5E-09	0.003

Further analysis of the SH3BP4 methylation in individuals with diabetes versus those without diabetes using linear mixed models and adjusting for the same covariates as in EWAS, revealed that the effect sizes were almost identical between the two groups (see figure below). This suggests that the relationship between mtDNA quantity and SH3BP4 methylation is not dependent on the diabetes status of an individual. The intercept in the group of twins with diabetes was lower than that of the twins without diabetes, indicating that, on average, twins with diabetes have lower mtDNA quantity compared to twins without diabetes. This supports our suggestion that diabetes represents the extreme end of obesity-related metabolic phenotypes.

4. Volcano Plots and CpG Sites: On page 4, lines 118-130, the volcano plots do not clearly show how mtDNA quantity is associated with methylation data. Only two CpG sites are significantly regulated, which is unexpectedly low. The authors should provide correlation plots to better visualize these associations and explain why so few CpG sites are regulated. Is this due to poor statistical power or other factors?

See Reviewer #3 Question 5.

5. Introduction of ICE FALCON: When introducing ICE FALCON in the results, the authors should provide a more detailed explanation of this model. A few sentences describing the model would help readers better understand its application and significance.

As the reviewer suggested, we added a brief explanatory sentence to describe the ICE FALCON method in the Results-section (pg. 7-8, lines 208-210).

6. Normalization Factors for mtDNA: On page 14, the authors chose APP and B2M as normalization factors for mtDNA quantity. This choice needs justification, as it is unclear why these factors were selected and whether they are present in muscle and adipose tissue. The authors should explain their assumptions and the rationale

behind this choice.

See Reviewer #3 Question 9.

Additionally, some minor points need to be addressed:

1. Abstract: Lines 49-51 need rewording for clarity.

We have reworded this sentence.

2. Results: On page 3, line 106, define DZ as dizygotic since it is the first mention.

We have modified this part.

3. Results: On page 3, line 109, remove "with."

See Reviewer #3 Question 4.

4. Results: On page 5, line 143, remove "in the."

See Reviewer #3 Question 6.

5. Results: On page 7, lines 191-193, provide the exact names for each item in Figure 4 for clarity.

See Reviewer #3 Question 7.

6. Figure Reference: There is no reference to Figure 5a in the paper; this needs to be included.

We have now added references for Figure 5a (pg. 8, line 210; pg. 19, line 513).

References

Castellani, C. A. *et al.* Mitochondrial DNA copy number can influence mortality and cardiovascular disease via methylation of nuclear DNA CpGs. *Genome Med.* **12**, 84 (2020).

Liang, K.-Y. & Zeger, S. L. Longitudinal Data Analysis Using Generalized Linear Models. *Biometrika* **73**, 13–22 (1986).

Distribution of obesity-related variables and mitochondrial DNA quantity

We are grateful for the opportunity to further revise the manuscript and would like to express our sincere thanks to the reviewers for their important and helpful insights, which have significantly improved the manuscript. We have addressed each of the comments in the sections below and have implemented validation analyses in the manuscript, along with other minor changes highlighted in yellow. Sections that have been moved within the manuscript, rather than newly added, are highlighted in blue.

Reviewer's Comments:

Reviewer #1 (Remarks to the Author)

The authors have responded the comments. I still have several comments as follows:

In this study, all twins were treated as individual subjects in EWAS analysis, whereas the ICE FALCON analysis was performed using MZ and DZ twins. The twin-specific characteristics were only considered in the ICE FALCON analysis, not in the EWAS. I recommend the authors conduct the EWAS analysis using twin models to better control for genetic and environmental influences.

We appreciate the reviewer's suggestion regarding the use of twins as individuals or conducting twin-specific analyses. We acknowledge that within-pair models (i.e., comparing the difference in methylation between twin A and twin B with the difference in mtDNAq between twin A and twin B) can effectively eliminate the effects of shared environmental and genetic factors. However, we believe that our individual-level EWAS aligns well with the ICE FALCON analyses. In both methods, we include family ID as a random effect, accounting for the higher correlation between the co-twins in each pair compared to other twins due to shared genetic and environmental factors. The ICE FALCON model, although applied to twins, is not a twin-specific model per se; it does not utilize within-pair data and can be applied to other related individuals. One scenario in ICE FALCON is the presence of familial confounding, i.e., shared environmental and genetic factors. As suggested by the reviewer, we repeated the association analysis between DNA methylation at *cg1999400* and mitochondrial DNA quantity using within-pair study design. We observed a similar effect size to the individual-level model (effect size -1.25 in the within-pair model vs. -1.27 in the individual-level model). This indicates that even after controlling for shared within twin pair factors, the effect remains, which is in line with our ICE FALCON findings that suggested causation rather than presence of familial confounding.

Reviewer #2 (Remarks to the Author)

After reviewing the authors' responses to the initial comments, I acknowledge the efforts made to address several key points raised. The authors emphasize the novelty of their work, particularly in linking DNA methylation, mitochondrial metabolism, and obesity across two relevant tissues—adipose and muscle. As such using multi-tissue approach and the potential modeling of causal pathways between associations is a valuable contribution. However, some of the initial concerns remain only partially addressed. While the authors highlight their findings in adipose tissue as evidence of tissue-specificity, however such an unexpectedly low association (one

CpG only) and absence of significant results in muscle tissue continues to raise questions about the robustness of these associations. Specifically when there is not validation strategy in place with respect to identified gene “SH3BP4”. Additionally, although the authors have mentioned modeling causal pathways, the underlying biological mechanisms still require further elucidation, which they acknowledge is a future research direction. The sample size remains a limitation, potentially affecting the power to detect broader tissue-specific effects and mechanistic insights. In conclusion, while the authors have made important clarifications and defended the novelty of their study, further work (e.g. validation / functional analysis with respect to SH3BP4 gene in the context of obesity) is needed to address the biological mechanisms, which would strengthen the overall impact of the findings.

We thank the reviewer for their comments and the important remarks that need addressing. Regarding the low number of significant CpGs, we believe this is related to the sample size, which remains a limitation of the study, as we have transparently noted in the manuscript. We are not suggesting that there are no other associations to be discovered. However, it is expected that some associations with likely small effect sizes cannot be reliably determined due to the modest sample size and the strict p-value threshold applied here to minimize false positive findings. This is why, instead of comparing only p-values, we used effect sizes to compare whether the direction or magnitude of the effects are similar between the two tissues. We believe this approach can still reveal trends in whether the overall DNA methylation profiles are similarly linked to mitochondrial DNA quantity (mtDNAq) across the tissues. Our data suggests that, in general, the relationship between DNA methylation and mtDNAq is different between the tissues in the context of obesity. Of course, this warrants further research and larger datasets to determine the biology underlying this observation. For instance, it may be that mtDNAq is not measuring the exact same mitochondrial function between the tissues. However, we would also like to note that tissue-specific effects are not the main focus of the paper, but rather an additional point we wanted to make due to having both adipose and muscle tissue data from the same individuals.

Regarding the *SH3BP4* methylation at *cg19998400* that we found to be linked with both adipose tissue mtDNAq and obesity-related variables, we fully agree that validating our results in an independent dataset is crucial for demonstrating the robustness of the findings. Therefore, we have now replicated our main findings in two independent European datasets: TwinsUK and the Scandinavian type 2 diabetes (T2D)-discordant MZ cohort. We have added paragraphs in our manuscript describing the replication cohorts (p. 21-22, lines 576-599) and the obtained results (p. 7, lines 194-206; p.8, lines 227-242). To summarize, we replicated the inverse association between *SH3BP4* methylation at *cg19998400* and mtDNAq in adipose tissue using the Scandinavian twin cohort. With the data from TwinsUK, we replicated the significant positive association between *SH3BP4* methylation and expression with BMI. A similar analysis using the Scandinavian T2D-discordant MZ twin cohort produced effects in the same direction but did not reach the statistical significance threshold of $p < 0.05$. Adjusting for cigarette smoking substantially increased the effect size in each of the analysis but reduced the sample size due to missing smoking data (please refer to Supplementary Table 9.). These discrepant results may be attributed to the low sample size or the high prevalence of T2D among participants, as discussed in the manuscript (p. 14, lines 381-396).

Reviewer #3 (Remarks to the Author)

The authors have addressed all my points, and the only remaining issues I see are:

1) ideally the links to genes with associations like SH3BP4 should have more links to other published work, or even if possible a replication cohort. Were they able to show any evidence of this from other groups?

We thank the reviewer for highlighting the importance of a validation cohort. We have now replicated our findings in two additional cohorts. Please refer to our response to Reviewer #2, where we describe the replication analysis in more detail.

2) All the code, methods, and scripts need to be clearly organized and public. The link they provided, <https://doi.org/10.6084/m9.figshare.26941927.v1>, only has the R session and two text files, but where is the raw data? Perhaps I missed it, but this should also be present and links to the code page.

As we state under “Data Availability” the FTC data is not publicly available due to restrictions in informed consent. In case the reviewer wishes to get access to the data, we ask them to contact the Institute for Molecular Medicine Finland (FIMM) Data Access Committee (DAC) (fimmdac@helsinki.fi) with IRB/ethics approval and an institutionally approved study plan.

Reviewer #4 (Remarks to the Author)

The authors have addressed all if my concerns and I believe should be ready for publication.

We are pleased to have addressed all the reviewer's concerns and would like to express our gratitude for their time and effort in reviewing our manuscript.